# ENHANCING NEURAL TRAINING VIA A CORRELATED DYNAMICS MODEL

**Jonathan Brokman**[*†]
jonathanbrok@gmail.com

**Roy Betser**[*†]
roybe@campus.technion.ac.il

**Rotem Turjeman**[*†]
rotem4004@gmail.com

**Tom Berkov**[†]
ptomberk@gmail.com

**Ido Cohen**[‡]
idocoh@ariel.ac.il

**Guy Gilboa**[†]
guy.gilboa@ee.technion.ac.il

## ABSTRACT

As neural networks grow in scale, their training becomes both computationally demanding and rich in dynamics. Amidst the flourishing interest in these training dynamics, we present a novel observation: Parameters during training exhibit intrinsic correlations over time. Capitalizing on this, we introduce *correlation mode decomposition* (CMD). This algorithm clusters the parameter space into groups, termed modes, that display synchronized behavior across epochs. This enables CMD to efficiently represent the training dynamics of complex networks, like ResNets and Transformers, using only a few modes. Moreover, test set generalization is enhanced.

We introduce an efficient CMD variant, designed to run concurrently with training. Our experiments indicate that CMD surpasses the state-of-the-art method for compactly modeled dynamics on image classification. Our modeling can improve training efficiency and lower communication overhead, as shown by our preliminary experiments in the context of federated learning.

## 1 INTRODUCTION

The research of training dynamics in neural networks aims to comprehend the evolution of model weights during the training process. Understanding stochastic gradient descent (SGD) is of paramount importance as it is the preferred training method used in practice. By studying these dynamics, researchers strive to achieve predictability, enhance training efficiency, and improve overall performance. In this work we show, for instance, how a good model can improve distributive and federated learning.

Outside the scope of deep learning, time profiles and correlation methods have been valuable for studying complex systems since the 1990s, as demonstrated by the foundational work of Aubry et al. (1991; 1992). Studies in variational image-processing such as Burger et al. (2016) and Gilboa (2018) also highlight the emergence of specific time profiles during gradient descent. However, these methods often rely on properties like convexity and homogeneity, which neural networks training dynamics typically lack.

Research into deep learning training dynamics dates back to the early days of neural networks, with initial studies focusing on two-layer models (Saad & Solla, 1995b;a; Mace & Coolen, 1998; Chizat & Bach, 2018; Aubin et al., 2018; Goldt et al., 2019; Oostwal et al., 2021). These simpler models continue to offer valuable insights, but our interest lies in the dynamics of contemporary, complex models that feature significantly larger numbers of weights and layers. These complexities introduce higher degrees of nonlinearity. Another popular simplification is the first-order Taylor approximation of the dynamics, particularly in the realm of infinitely wide neural networks, as in Jacot et al. (2018); Lee et al. (2019). However, this often results in performance shortcomings.

A different approach examines the low-dimensional sub-space hypothesis, positing that effective training can occur in a limited set of directions within the weight space, see Li et al. (2018a); Gur-Ari et al. (2018); Gressmann et al. (2020). Recently, Li et al. (2023) employed the eigenvectors

---

[*]The contribution of these researchers to this work is equal.

[†]Technion – Israel Institute of Technology,[‡]Ariel University

of the covariance between weight trajectories to define a 40-dimensional training sub-space. They achieve strong performance in this constrained sub-space, setting the SOTA for compactly modeled dynamics in image classification. Other studies proposed to represent the training dynamics of complex models through a concise set of representative trajectories $w_m(t)$:

$$w_i(t) \approx b_i + \sum_m a_{i,m} w_m(t), \tag{1}$$

where $b_i, a_{i,m}$ are scalars and $w_i(t)$ is the trajectory of the $i$'th tunable weight (parameter) of the network. Common solution approaches employ dynamical analysis techniques, relying on Koopman theory and Dynamic Mode Decomposition (DMD), see Schmid (2010); Dogra & Redman (2020); Naiman & Azencot (2021). Our experiments and analysis indicate that DMD and its variants do not adequately capture the complex dynamics inherent in neural network training (see Fig. 1 and Appendix H).

Our main observation is that network parameters are highly correlated and can be grouped into "Modes" characterized by their alignment with a common evolutionary profile. This captures the complex dynamics, as illustrated in Fig. 1. Our findings are substantiated through comprehensive experiments; a selection of key graphs is presented in the main body, while the Appendix offers a rich array of additional results, ablation studies, stability evaluations and details regarding various architectures and settings. Our contributions are three-fold:

- The introduction of Correlation Mode Decomposition (CMD), a data-driven approach that efficiently models training dynamics.

- Devising efficient CMD which can be performed online during training, out-performing the state-of-the-art method for training dynamics dimensionality reduction of Li et al. (2023) in both efficiency and performance.

- We pioneer the use of modeled training dynamics to reduce communication overhead. Our preliminary experiments offer a promising avenue for future research.

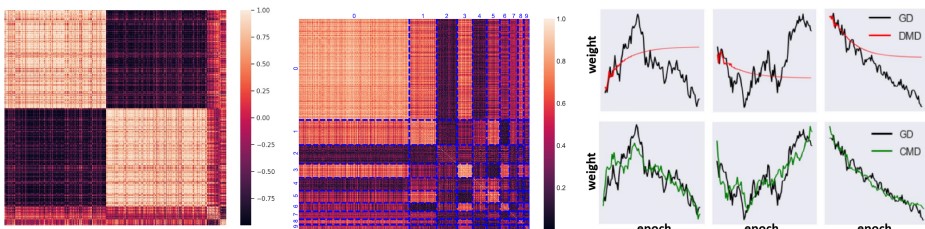

Figure 1: **Correlated dynamics**. Left: Correlation matrix of weight trajectories, clustered to modes, for a simple network (See Fig. 10a in the Appendix) trained on MNIST (LeCun et al. (1998)). Middle: Clustered correlation matrix of weight trajectories for ResNet18 (He et al. (2016a)) trained on CIFAR10 (Krizhevsky et al. (2009)). The block diagonal structure indicates high correlations of the parameter dynamics within each mode. Right: Three weight trajectories from SGD training are reconstructed via DMD, which uses exponentials (Top) and CMD, which uses reference trajectories (Bottom). These highlight the effectiveness of CMD in capturing complex training dynamics. See Appendix A.1, Figs. 6, 7 for extended findings.

## 2 PREVIOUS WORK

Dimensionality reduction techniques, particularly those using correlation matrices, are key to our approach and have roots in statistics and information visualization, see Nonato & Aupetit (2018); Sorzano et al. (2014); Van Der Maaten et al. (2009); Van der Maaten & Hinton (2008). Various studies explore low-dimensional sub-space training using similar methods. For example, random projections can capture a substantial amount of the gradient's information (Li et al., 2018a). Hessian-based methods focus on its top eigenvectors, (Gur-Ari et al., 2018). Dimensionality has been further reduced via smart weight sampling, (Gressmann et al., 2020). The work by Li et al. (2023) bears the most resemblance to our research. The authors affirm low-dimensional training dynamics, using top

eigenvectors of the covariance matrix to span the training space. The gradient is fully computed for all of the weights before being projected onto this basis, making their method more involved than regular (stochastic) gradient descent. They also introduce an efficient Quasi-Newton training scheme that benefits from the low-dimensional representation. Our approach distinguishes itself from Li et al. (2023) by focusing on dynamics mode decomposition, rather than training in a low-dimensional sub-space. Both methods use dimensionality reduction to compactly model the dynamics, but operate differently. Our analysis is not based on eigenvectors but on nonlinear clustering, yielding improved performance. DMD and Koopman theory, (Schmid, 2010; Tu, 2013; Williams et al., 2015; Kutz et al., 2016), decompose dynamics to modes using well-studied operators. They extend to neural training dynamics analysis (Manojlović et al., 2020; Dogra & Redman, 2020; Redman et al., 2021), including advanced methods for highly non-linear cases Yeh et al. (2023). Šimánek et al. (2022) also incorporates DMD for training optimization. Contrary to these, we extract modes from the observed correlation in the dynamics without using a predefined operator, implying a more adaptive, data-driven approach. The geometry of loss landscapes has been a topic of interest, see Shalev-Shwartz & Ben-David (2014); Keskar et al. (2016); Li et al. (2018b); Draxler et al. (2018). We showcase our dimensionality-reduction technique in this regard as well.

We demonstrate the relevance of our approach for communication overhead in distributed learning, a widely-studied challenge addressed by various techniques, including order scheduling and data compression (Ho et al., 2013; Cui et al., 2014; Zhang et al., 2017; Chen et al., 2019; Shi et al., 2019; Hashemi et al., 2019; Peng et al., 2019; Dryden et al., 2016; Alistarh et al., 2017). In Federated Learning, communication is particularly problematic (Li et al., 2020). For instance, Chen et al. (2021) finds communication takes up more than 50% of the training time in their experiments. Solutions involve adaptive weight updates to reduce communication load (Seide et al., 2014; Alistarh et al., 2017; Hsieh et al., 2017; Luping et al., 2019; Konečnỳ et al., 2016; McMahan et al., 2017; Ström, 2015; Chen et al., 2021). In our work we propose for the first time to use dynamics modeling to reduce the number of communicated parameters.

## 3 METHOD

We present several approaches to apply our method, starting by discussing the underlying correlated dynamics (Sec. 3.1). Following this discussion we introduce the basic CMD version, called *post-hoc* CMD, which analyzes dynamics after they occur (Sec. 3.2). We then present the more efficient *online* CMD (Sec. 3.3) and *embedded* CMD (Sec. 3.4), which embeds CMD dynamics to the training process. In Sec. 3.5 we demonstrate accuracy landscape visualization via CMD.

### 3.1 THE UNDERLYING CORRELATED DYNAMICS

**Hypothesis 1** *The dynamics of neural network parameters can be clustered into very few highly correlated groups, referred to as "Modes".*

Let $\mathbf{W} = \{w_i(t)\}_{i=1}^N$ denote the set of time-profiles of the neural network trainable weights during the training process, where $i$ is the weight index, $t$ is the training time, and $N$ is the total number of such weights. We refer to $\mathbf{W}$ as the 'trajectories' where each $w_i(t)$ is a single 'trajectory'. The trajectories are clustered to $M$ modes, where $M \ll N$. Let $\mathbf{W}_m \subset \mathbf{W}$ be the subset of trajectories corresponding to mode $m$ within $\mathbf{W}$. Using a minor abuse of notation, when $t$ is discrete we use $\mathbf{W}, \mathbf{W}_m$ to denote the matrices where the (discrete) trajectories are arranged as rows. Details of the discrete case will be provided in Sec. 3.2. The correlation of two trajectories $u, v$ is denoted $corr(u, v)$. Any two time trajectories $u, v$ that are perfectly correlated (or anti-correlated), i.e. $|corr(u, v)| = 1$, can be expressed as an affine transformation of each other, $u = av + b$. Thus, under the assumption of correlated dynamics, the following approximation is used to describe the dynamics of a weight trajectory

$$w_i(t) \approx a_i w_m(t) + b_i, \ \forall w_i(t) \in \mathbf{W}_m, \tag{2}$$

where $a_i$, $b_i$ are scalar affine coefficients associated with $w_i$, and $w_m(t)$ is a single common weight trajectory chosen to represent mode $m$ termed 'reference weight'. Note that in terms of the usual dynamics decomposition approach of Eq. (1), the case of Eq. (2) is a specific case where a single $w_m(t)$ models each trajectory. Figs. 1, 7 demonstrate that Eq. (2) approximates the dynamics using

a notably small set of modes (10 or less in most cases), supporting our hypothesis of correlated dynamics. Surprisingly, parameters of the same layers are typically distributed between the modes, see Fig. 10b in the Appendix.

While neural networks are notoriously known for their complex dynamics, the data-driven approach of Eq. (2) is simple yet effective at capturing these training intricacies. The reference weights hold the advantage of not being restricted to common assumptions typically made in classical dynamics analysis. For instance, we show DMD (a classical analysis approach) to be too restrictive even for simple cases in Appendix H.2.

Our key novelty lies in leveraging the correlation between parameter trajectories to compactly model the training dynamics. We will show that this approach outperforms current modeling methods.

### 3.2 POST-HOC CMD

In this section we introduce the basic CMD version, called *post-hoc* CMD, which analyzes dynamics after they occur. This algorithm models dynamics as per Eq. (2): It splits the trajectories to modes, associates a reference trajectory to each mode and approximates the affine coefficients.

Consider a discrete time setting, $t \in [1, ..E]$, where $E$ is the number of epochs, and denote $N_m$ the number of trajectories in mode $m$. Then $w_m, w_i \in \mathbb{R}^E$, $\mathbf{W} \in \mathbb{R}^{N \times E}$, $\mathbf{W}_m \in \mathbb{R}^{N_m \times E}$. We consider 1D vectors such as $u \in \mathbb{R}^E$ to be row vectors, and define $\bar{u} := u - \frac{1}{E} \sum_{t=1}^{E} u(t)$, $\langle u, v \rangle = uv^T$, $\|u\| = \sqrt{uu^T}$, $corr(u, v) = \frac{\langle \bar{u}, \bar{v} \rangle}{\|\bar{u}\|\|\bar{v}\|}$. Let $A, B \in \mathbb{R}^N$ s.t. $A(i) = a_i$, $B(i) = b_i$, and let $A_m, B_m \in \mathbb{R}^{N_m}$ be the parts of $A, B$ corresponding to the weights in mode $m$. Denote $\tilde{w}_m = \begin{bmatrix} w_m \\ \vec{1} \end{bmatrix} \in \mathbb{R}^{2 \times E}$, where $\vec{1} = (1, 1, \cdots 1) \in \mathbb{R}^E$, and $\tilde{A}_m = [A_m^T, B_m^T] \in \mathbb{R}^{N_m \times 2}$. In this context, Eq. (2) reads as

$$\mathbf{W}_m \approx \tilde{A}_m \tilde{w}_m. \tag{3}$$

**Step 1. Find reference trajectories.** Given the trajectories $w_i \in \mathbf{W}$, use correlation to select $w_m$. To avoid a large $N \times N$ matrix, we sample $K$ trajectories to construct the absolute trajectory correlation matrix $C_1$, i.e.

$$C_1[k_1, k_2] = |corr(w_{k_1}, w_{k_2})|, C_1 \in \mathbb{R}^{K \times K}, \tag{4}$$

where $w_{k_1}, w_{k_2}$ take any pair of the $K$ sampled trajectories and $M \ll K \ll N$. $C_1$ is used to cluster the $K$ trajectories to $M$ modes. In each mode, $w_m$ is chosen as the weight trajectory with the highest average correlation to the other weights in the mode. Details on the sampling and clustering procedures are available in Appendix C.2, C.3.

**Step 2. Associate weights to modes.** Given $\{w_m\}_{m=1}^M$ and $\mathbf{W}$, partition $\mathbf{W}$ to $\{\mathbf{W}_m\}_{m=1}^M$. Let

$$C_2 \in \mathbb{R}^{N \times M} : C_2[i, m] = |corr(w_i, w_m)|. \tag{5}$$

To get $\{\mathbf{W}_m\}_{m=1}^M$, we obtain the mode of each $w_i \in \mathbf{W}$ as $\arg\max$ of the $i^{th}$ row $C_2[i, :]$.

**Step 3. Obtain the affine parameters.** Given $w_m, \mathbf{W}_m$, get $\tilde{A}_m$ by a pseudo-inverse of Eq. (3)

$$\tilde{A}_m = \mathbf{W}_m \tilde{w}_m^T (\tilde{w}_m \tilde{w}_m^T)^{-1} \in \arg\min_{\tilde{A}_m} \|\mathbf{W}_m - \tilde{A}_m \tilde{w}_m\|_F, \tag{6}$$

where $\| \cdot \|_F$ denotes the Frobenius norm. Reconstruction of the modeled dynamics $\mathbf{W}_m^{Recon}$, is performed by re-plugging $\tilde{A}_m$ to the right-hand-side of Eq. (3)

$$\mathbf{W}_m^{Recon} = A_m^T w_m + B_m^T \vec{1}, \tag{7}$$

where, e.g., mode $m$ weights at end of training appear at the last column, denoted $\mathbf{W}_m^{Recon}(t)\Big|_{t=E}$.

We tested post-hoc CMD on image classification, segmentation and generative image style transfer. The performance upon replacing the weights with the CMD modelled weights produces similar, and even enhanced results, compared to the original network weights, see Fig. 2. Further experimental details and additional post-hoc results, and ablation studies are in Appendix B. As far as we know, we are the first to assess dynamics modeling in image segmentation and generative style transfer. Post-hoc CMD is resource-intensive due to full training trajectory usage. Thus we introduce efficient CMD variants for real-time computation during training.

Figure 2: **Post-hoc CMD examples**. CMD representation is highly general and models well diverse architectures operating on various tasks. Here, the successful modeling is evident through its ability to follow the performance dynamics while even providing a performance boost. Left: CIFAR10 performance on ViT-b-16 (Dosovitskiy et al., 2020), pre-trained on JFT-300M. Middle: StarGAN-v2 (Choi et al., 2020) style transfer, qualitative result of the original network compared to the result of CMD modeling of its dynamics. Right: Segmentation results, using PSPNet Architecture (Zhao et al., 2017). See more details and results in Appendix B and Figs. 15, 18. 'GD' in the legend stands for SGD.

## 3.3 ONLINE CMD

We introduce an iterative CMD version that mitigates the computational drawbacks of post-hoc CMD. Designed to run alongside the regular training process, with negligible overhead - we term it *online*. Transforming post-hoc CMD into an iterative algorithm, we eliminate the need for full training trajectories, reducing memory and computation.

**Warm-up phase.** We observed that the reference trajectories can be selected once during training, and do not require further updates (see Fig. 16 and Table 4 in Appendix C.5). Thus we perform Step 1 of Sec. 3.2 once at a pre-defined epoch $F$, and the selected reference weights continue to model the dynamics for the remainder of the training process. After the warm-up phase, we perform efficient iterative updates, variants of Step 2 and Step 3 of Sec. 3.2.

**Step 2 (iterative).** This step requires computing $C_2$, the trajectory correlation matrix of Eq. (5). $C_2$ is calculated using inner products and norms. Denote $w_t \in \mathbb{R}^t$ as the discrete trajectory of a weight $w$ at times 1 through $t$. Given $\langle w_{m,t-1}, w_{i,t-1} \rangle, \|w_{i,t-1}\|^2$, we may obtain $\langle w_{m,t}, w_{i,t} \rangle, \|w_{i,t}\|^2$ as:

$$\langle w_{m,t}, w_{i,t} \rangle = \langle w_{m,t-1}, w_{i,t-1} \rangle + w_m(t)w_i(t), \quad \|w_{i,t}\|^2 = \|w_{i,t-1}\|^2 + w_i^2(t). \tag{8}$$

**Step 3 (iterative).** Let $\tilde{A}_m(t)$ be $\tilde{A}_m$ evaluated on trajectories up to time $t$. Given $\tilde{A}_m(t-1)$, we have by Eq. (6)

$$\tilde{A}_m(t) = \left( \tilde{A}_m(t-1)(\tilde{w}_{m,t-1}\tilde{w}_{m,t-1}^T) + \mathbf{W}_m(t)\tilde{w}_m^T(t) \right) (\tilde{w}_{m,t}\tilde{w}_{m,t}^T)^{-1}, \tag{9}$$

$$\tilde{w}_{m,t}\tilde{w}_{m,t}^T = \tilde{w}_{m,t-1}\tilde{w}_{m,t-1}^T + \begin{pmatrix} w_m^2(t) & w_m(t) \\ w_m(t) & 1 \end{pmatrix}, \tag{10}$$

where $\mathbf{W}_m(t), \tilde{w}_m(t)$ are the $t^{th}$ column of $\mathbf{W}_m, \tilde{w}_m$ respectively, and $\tilde{w}_{m,t}$ are columns 1 through $t$ of $\tilde{w}_m$. Note that we only used the weights at the current time $t$, along with the $2 \times 2$ matrices $\tilde{w}_{m,t-1}\tilde{w}_{m,t-1}^T, \tilde{w}_{m,t}\tilde{w}_{m,t}^T$, which are also iteratively maintained via Eq. (10). In all of our experiments - invertability of this $2 \times 2$ was never an issue (generally, a pseudo-inverse can be used). Thus we use Eq. (9), Eq. (10) for iterative updates, see derivation of Eq. (9) in Appendix C.4.

**Performance** Our experiments show that after the $F$ warm-up iterations we may fix not only Steps 1, but also Step 2, and continue with iterative Step 3 updates, while maintaining on-par performance with post-hoc CMD, as shown in Table 4 in Appendix C.5. Online CMD outperforms the current SOTA low-dimensional model of Li et al. (2023) on various image classification architectures, as highlighted in Table 1, while requiring a substantially shorter warm-up phase.

**Efficiency** Unlike Eq. (6), which is performed once at the end of the warm-up phase, the iterative updates, Eq. (9), Eq. (10), do not require trajectory storage at all. Moreover, their overhead is computationally negligible compared to a training step, see Appendix C.6 for the calculations. Thus we have an online dynamics modeling procedure with negligible overhead - setting the ground for training enhancement. The full Online CMD algorithm is available in Algorithm 3 in Appendix C.1.

### 3.4 EMBEDDING THE MODELED DYNAMICS IN TRAINING

This method gradually integrates CMD dynamics into training, aimed at leveraging modeled dynamics for enhanced learning. This improves accuracy and reduces the number of trained parameters. Experiments show that gradual CMD embedding stabilizes training, whereas instantaneous CMD use in SGD is unstable (Fig. 3). Two naive approaches failed: 1) periodic CMD weight assignment reverted to regular SGD paths; 2) training only the reference weights after the warm-up phase.

Let us describe the gradual embedding approach of choice. We rely upon the online algorithm of Sec. 3.3, where after a short warm-up phase the affine parameters $a_i, b_i$ are iteratively updated at each epoch. Every $L$ epochs we further embed additional parameters. A parameter $w_i$ for which the model is stable (change of $a_i, b_i$ in time is negligible) is CMD embedded from then on, i.e. it gets its value according to the CMD model and not by gradient updates. Additionally, from this point on, $a_i, b_i$ are fixed.

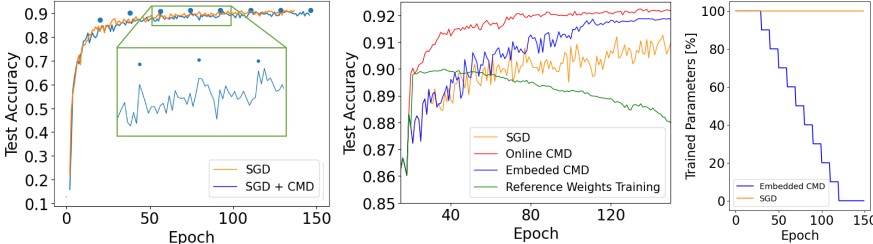

Figure 3: **Embedding CMD in training**. Left: Naively assigning CMD-modeled weights to the model every 20 epochs, and performing SGD in between. Dots represent the CMD model performance. Performance is repeatedly drawn back to the CMD-less (regular SGD) case. Middle: Online CMD and Embedded CMD, compared to regular SGD. An additional naive approach which trains only the reference weights post warm-up is also presented (green). Performance is portrayed along the training epochs. CMD runs use $M = 10$ modes, $F = 20$ warm-up epochs. We use an embedding rate of $P = 10\%$, and $L = 10$ epochs (Right), which lead to $\approx 50\%$ reduction in the total number of trained parameters in the whole training cycle. Both online and embedded CMD variants considerably improve performance, compared to regular SGD. Both naive approaches do not.

As shown in Appendix C.8, the affine parameters $a_i, b_i$ tend to exhibit minimal or no changes after a certain epoch, denoted $t = \tau_i$. Thus, after $\tau_i$ we may fix $a_i, b_i$, i.e. use $a_i \leftarrow a_i(\tau_i), b_i \leftarrow b(\tau_i) \forall t > \tau_i$. Denote $\mathcal{I}(t) = \{i : t > \tau_i\}$. To embed the modeled dynamics to $w_i$, we perform

$$w_i(t) \leftarrow \begin{cases} \text{SGD update} & \text{if } i \notin \mathcal{I}(t) \\ a_i w_m(t) + b_i & \text{if } i \in \mathcal{I}(t) \end{cases}. \tag{11}$$

After $\tau_i$, we say that $w_i$ is an *embedded weight*, and $a_i, b_i$ its *embedded coefficents*. Note: Embedded coefficients are frozen in time, embedded weights are not. Let $c_i$ be a criterion of change in $a_i, b_i$, for instance $c_i(t) = \sqrt{\|a_i(t) - a_i(t-L)\|^2 + \|b_i(t) - b_i(t-L)\|^2}$. One way to obtain $\tau_i$ is by

$$\tau_i = t : c_i(t) \text{ is in the bottom P\% of the unembedded weights.} \tag{12}$$

Other reasonable ways include $\tau_i = t : c_i(t) < \epsilon$, for un-embedded $w_i$, using a small threshold $\epsilon > 0$. The full online embedding algorithm is available in Algorithm 4 in the Appendix. Note that we do not embed the reference weights $\{w_m\}_{m=1}^M$. Sec. 4 demonstrates the benefit of embedded dynamics in the context of federated learning, where we can locally store the embedded coefficients to significantly reduce the communication overhead.

### 3.5 CMD ACCURACY AND LOSS LANDSCAPE VISUALIZATION

Reducing the search space to 1-2 dimensions allows for loss landscape visualization. To this end we use CMD with 2 modes, as seen in Fig. 4, where different parts of the dataset are used for accuracy visualization. Expectedly, training and validation sets differ in landscape smoothness. Smoothness as well as minima points also vary across the CIFAR10 classes. See further details in Appendix G.

Figure 4: **Visualization of accuracy landscape**. A grid is created based on the two reference weights, and colored by accuracy values. The accuracy landscape of the training set (left) is very smooth, compared to that of the validation set (second left). The landscape of the Automobile class (3rd left) is rather regular whereas the Dog class (right) is much more irregular and suboptimal. The original CMD model is marked with a black dot, the optimal model is marked with a red dot.

# 4 CMD FOR EFFICIENT COMMUNICATION IN DISTRIBUTED LEARNING

We propose for the first time to use training dynamics modeling for efficient communication in distributed learning. In conventional (non-distributed) learning, as discussed above, a reduction in trained parameters leads to decreased computation. Nonetheless, in distributed scenarios, modeling trajectories may reduce the No. of communicated parameters as well, offering further significant benefits. Specifically in Federated Learning (FL), communication efficiency is crucial (Li et al., 2020). For example, Chen et al. (2021) find in their FL experiments that communication accounts for over 50% of the total training time. Following their experimental setting, we concentrate on FL, where client models handle unique per-client data. Although developed for FL, our method is adaptable for distributed learning.

Let $\mathcal{C}$ be the set of client models, trained locally on distinct data. Periodical 'synchronization rounds' occur at time $t_s$ as follows: A subset $C(t_s) \subset \mathcal{C}$ is randomly selected and aggregated in a Central Server using standard averaging (McMahan et al., 2017). We consider the (less standard) practice that aggregates only a subset of the weights

$$\text{Central Server: } w_i^{\text{main}}(t_s) = \frac{1}{|C(t_s)|} \sum_{\text{client} \in C(t_s)} w_i^{\text{client}}(t_s), \, \forall i \notin \mathcal{I}(t_s), \tag{13}$$

where $\mathcal{I}(t_s)$ is the set of indices of the un-aggregated weights at time $t_s$. Un-aggregated weights do not require communication with the central server, hence increasing the size of $\mathcal{I}$ while maintaining performance is beneficial. To this end, we harness the embedded CMD modeling of Sec. 3.4, where the analyzed set of weight trajectories is $\{w_i^{\text{main}}(t_s)\}_{i \in \mathcal{I}(t_s)}$ as follows: The coefficients $\{a_i, b_i\}$ are maintained in the Central Server as in Sec. 3.4, and calculated w.r.t. $\{w_i^{\text{main}}(t_s)\}_{i=1}^N$. Let $\tau_{s,i}$ be the round in which $a_i, b_i$ are embedded. At $t_s = \tau_{s,i}$, the pair $\{a_i, b_i\}$ is distributed to the client models. Each client maintains a local copy of all the embedded coefficients $\{a_i, b_i\}_{i \in \mathcal{I}(t_s)}$. The client synchronization is performed, similarly to Eq. (11), as

$$\text{Local Client: } w_i^{\text{client}}(t_s) \leftarrow \begin{cases} w_i^{\text{main}}(t_s) & \text{if } i \notin \mathcal{I}(t_s) \\ a_i w_m(t_s) + b_i & \text{if } i \in \mathcal{I}(t_s), \end{cases} \tag{14}$$

where $\mathcal{I}(t_s) = \{i : t_s > \tau_{s,i}\}$, and $\tau_{s,i}$ is updated at the Central Server via Eq. (12). Note that only the weights corresponding to indices $i \notin \mathcal{I}(t_s)$ and the set $\{w_m\}_{m=1}^M$ require communication. Additionally, each $a_i, b_i$ pair is sent once during the training from the main model to the clients (when they are selected as embedded coefficients). Note: CMD is performed solely on $\{w_i^{\text{main}}(t_s)\}_{i=1}^N$.

In basic FL, $N$ weights from each of the $|C|$ selected clients are sent to a central server during each synchronization step. These are aggregated into a main model, which is then distributed back to all $|\mathcal{C}|$ clients. Thus, before employing any compression, the total data volume transmitted for each synchronization, is given by:

$$\text{Sync. Volume (Baseline)} = N(|C| + |\mathcal{C}|), \tag{15}$$

where usually this is measured in floating point numbers (depending on the variable-type of the $N$ weights).

In our scenario, due to the progressive change in $\mathcal{I}$, the communicated volume is not constant. Thus, we consider the following average:

$$\text{Sync. Volume (CMD)} = \hat{N}^{\text{not-embedded}}(|C| + |\mathcal{C}|) + \frac{2N}{E_s}|\mathcal{C}|, \qquad (16)$$

where $\hat{N}^{\text{not-embedded}}$ is the number of un-embedded parameters, averaged over $E_s$ synchronization rounds. Note that upon full embedding, the $M$ reference weights are still required for communication. The term $\frac{2N}{E_s}|\mathcal{C}|$ accounts for the following: Each $a_i$, $b_i$ pair is embedded and communicated once during training. The $N$ pairs form a total of $2N$ communicated parameters. Communication is performed to all clients, i.e. $2N|\mathcal{C}|$ total volume. Average round communication is thus $\frac{2N}{E_s}|\mathcal{C}|$.

In Fig. 5 our approach is compared to Chen et al. (2021), showing comparable accuracy with significant reduction of communicated weights.

## 5 RESULTS

We provide a focused comparison between our approach and the most closely related contemporary works: Li et al. (2023) that uses the SOTA P-BFGS for compactly modeled dynamics in image classification, and Chen et al. (2021)'s APF and aggressive APF (A-APF) aimed at reducing communicated parameters in FL. While we are first to leverage dynamics modeling for efficient communication, APF targets parameters that are nearly constant over time, which is infact plateau dynamics.

Table 1: **CIFAR10 test accuracy results for different modeling methods**. P-BFGS stands for the algorithm proposed in Li et al. (2023). Warm-up phase length (equal to CMD or longer) is noted in each P-BFGS column title. Best result in each row is in bold, second best is underlined. Vit-b-16 model is pre-trained on ImageNet1K. For the full experimental details see Appendix D, Tables 5, 6 specifically.

| Model | Baseline SGD | Online CMD | P-BFGS (equal) | P-BFGS (long) |
|---|---|---|---|---|
| ResNet18 | $90.91\% \pm 0.52$ | $\mathbf{92.43\%} \pm 0.34$ | $89.34\% \pm 0.54$ | $\underline{91.03\%} \pm 0.38$ |
| WideRes | $92.70\% \pm 0.48$ | $\mathbf{93.21\%} \pm 0.50$ | $88.65\% \pm 0.77$ | $\underline{92.78\%} \pm 0.25$ |
| PreRes164 | $90.64\% \pm 0.98$ | $\mathbf{91.34\%} \pm 1.20$ | $90.76\% \pm 1.14$ | $\underline{90.91\%} \pm 0.77$ |
| LeNet-5 | $75.36\% \pm 0.57$ | $\mathbf{76.77\%} \pm 0.60$ | $74.67\% \pm 0.35$ | $\underline{76.44\%} \pm 0.37$ |
| GoogleNet | $92.05\% \pm 0.48$ | $\mathbf{93.07\%} \pm 0.49$ | $92.01\% \pm 1.05$ | $\underline{92.41\%} \pm 0.38$ |
| ViT-b-16 | $\underline{97.86\%} \pm 0.11$ | $\mathbf{97.99\%} \pm 0.28$ | $95.05\% \pm 0.69$ | - |

**Dimensionality reduction.** In Table 1 we compare our online CMD method to P-BFGS (Li et al., 2023) on various architectures, ranging from very simple networks to vision transformers. We add results of CMD embedding for some of the models in Table 10 in the Appendix. For implementation details see Appendix D. P-BFGS appears twice in the table: With CMD-equivalent warm-up, and with an 80-epoch warm-up as they originally propose. Dimensionality is 40 unless the warm-up is ≤40 epochs, then it matches the warm-up length due to method constraints. Our online CMD method outperforms P-BFGS in all experiments. With just a handful of modes (10 or less), we achieve superior performance, regardless of the warm-up phase duration. P-BFGS requires substantially more dimensions and a longer warm-up phase than CMD, yielding models with low performance for equal warm-up phases to CMD and less than 40 dimensions. Furthermore, Online CMD has constant memory consumption and negligible computational overhead (see Appendix C.6, C.7). On the other hand, P-BFGS requires a matrix linearly dependent on the number of dimensions, leading to memory issues when training large models such as Wide-ResNet (Zagoruyko & Komodakis, 2016) or a vision transformer - ViT-b-16 (Dosovitskiy et al., 2020). In fact, we could not train P-BFGS on ViT-b-16 using a single GPU, like CMD. Moreover, we could not calculate the P-BFGS parameters for the long warm-up case using our computational device (125GB RAM memory), therefore only the equal warm-up phase option is available in the results.

**CMD as a regularizer** A by-product of applying CMD for dynamics modeling is smoother trajectories, demonstrated in Figs. 1, 7. Smoothed trajectories have been shown to improve performance,

for instance in using EMA and SWA, (Tarvainen & Valpola, 2017; Izmailov et al., 2018). We compare this performance boost with CMD in Table 2, using several SGD variants. It can be observed that CMD even slightly out-performs EMA and SWA. See additional information in Appendix D. Notably, EMA and SWA are specifically designed for smoothing and relate to CMD solely through this by-product; they do not involve dynamics modeling.

Table 2: **CIFAR10 test accuracy results on ResNet18 for different regularizers**. Different training optimization techniques are used. Best result in each row is in bold, second best is underlined. CMD entails excellent implicit regularization, competing well with dedicated training regularization algorithms. 'Mom' stands for momentum and 'SCH' stands for learning rate scheduler.

| Optimizer / Method | Baseline SGD | EMA | SWA | Online CMD |
|---|---|---|---|---|
| SGD | 90.68%±0.29 | 92.02%±0.16 | 92.09%±0.10 | **92.26%** ± 0.12 |
| SGD + Mom | 90.91%±0.52 | 92.22%±0.43 | 92.28%±0.44 | **92.43%** ± 0.34 |
| SGD + Mom + SCH | 91.29%±0.23 | 91.28%±0.23 | 91.29%±0.24 | **91.68%** ± 0.21 |
| Adam | 89.59%±0.43 | 91.43%±0.39 | 91.65%±0.43 | **91.70%** ± 0.51 |

**Federated Learning** In Table 3 we contrast CMD with APF and A-APF. As a baseline, we perform FL training without any communication boosting. The *Volume* criterion quantifies the average number of parameters communicated per round, expressed as a ratio to the baseline number of parameters. We note that no additional compression is performed here. We closely followed the experimental setting provided in Chen et al. (2021), as detailed in Appendix F.1. CMD is preferable in terms of *Volume* to APF and A-APF as seen in Table 3.

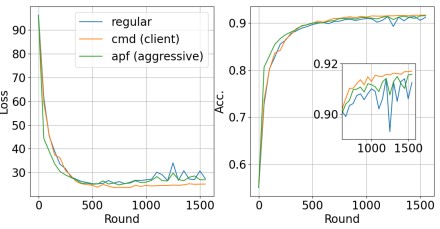

Figure 5: **Federated Learning, ResNet18 on CIFAR10**. Test loss and accuracy throughout training. CMD with two modes is harnessed for FL, compared to regular SGD (Baseline) and Aggressive APF (A-APF). Each trend-line is the average of 10 runs.

Table 3: **Federated Learning, ResNet18 on CIFAR10**. CMD significantly reduces volume (48.12%) while boosting test accuracy compared to the baseline. In contrast, APF marginally reduces communication but attains a slightly higher accuracy boost, and A-APF reduces communication (not as much as CMD) with a negligible accuracy boost. Overall, CMD offers a favorable trade-off between communication and accuracy.

| | Volume | Test Accuracy |
|---|---|---|
| Baseline | 100% | 91.4% ±0.3 |
| Ours | **48.12**% | 91.7% ±0.2 |
| APF | 90.22% | **92.0%** ±0.2 |
| A-APF | 60.75% | 91.5% ±0.4 |

## 6 CONCLUSION

In this paper, we have presented a new approach to model neural network parameter dynamics, leveraging the underlying correlated behavior of the training process. We show the model applies for a wide variety of architectures and learning tasks. By incorporating a low-dimensional approximation and using iterative update techniques, we introduce Online CMD, an efficient online method that models the training dynamics with just a handful of modes, while enhancing performance. We also incorporated the modeled dynamics into the training process and demonstrated the efficiency in Federated Learning, improving the communication overhead. The idea to use modeled dynamics for this task is novel in itself.

Our experiments showcase a range of benefits: We improve accuracy relative to regular SGD training and also surpass the state-of-the-art in low-dimensional dynamics modeling. The model can also be used effectively for visualization of accuracy and error landscapes. Additionally, our federated learning method significantly reduces communication overhead without compromising performance. Our approach could be effectively combined with other techniques, opening up new avenues for optimizing complex neural network training.

## REPRODUCIBILITY

Detailed pseudo-code of our CMD algorithm is described in Algorithms 1, 2. Our other CMD variants, Online CMD and Embedded CMD are descibed in Algorithms 3, 4. All the algorithms are available in the Appendix. The parameters used for our experiments in Sec. 5 are available in the Appendix D, specifically Table 5 (training parameters) and Table 6 (CMD parameters). All of our algorithms: Post-hoc, Online, and Embedded CMD, have an implementation provided here, alongside a script which executes them in the setting of the results in Sec. 5.

## ACKNOWLEDGEMENTS

GG would like to acknowledge support by the Israel Science Foundation (Grants No. 534/19 and 1472/23), by the Ministry of Science and Technology (Grant No. 5074/22) and by the Ollendorff Minerva Center.

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

# APPENDIX

## A  CORRELATION RELATED EXPERIMENTS

### A.1  MANUAL INSPECTION OF MODELED TRAJECTORIES

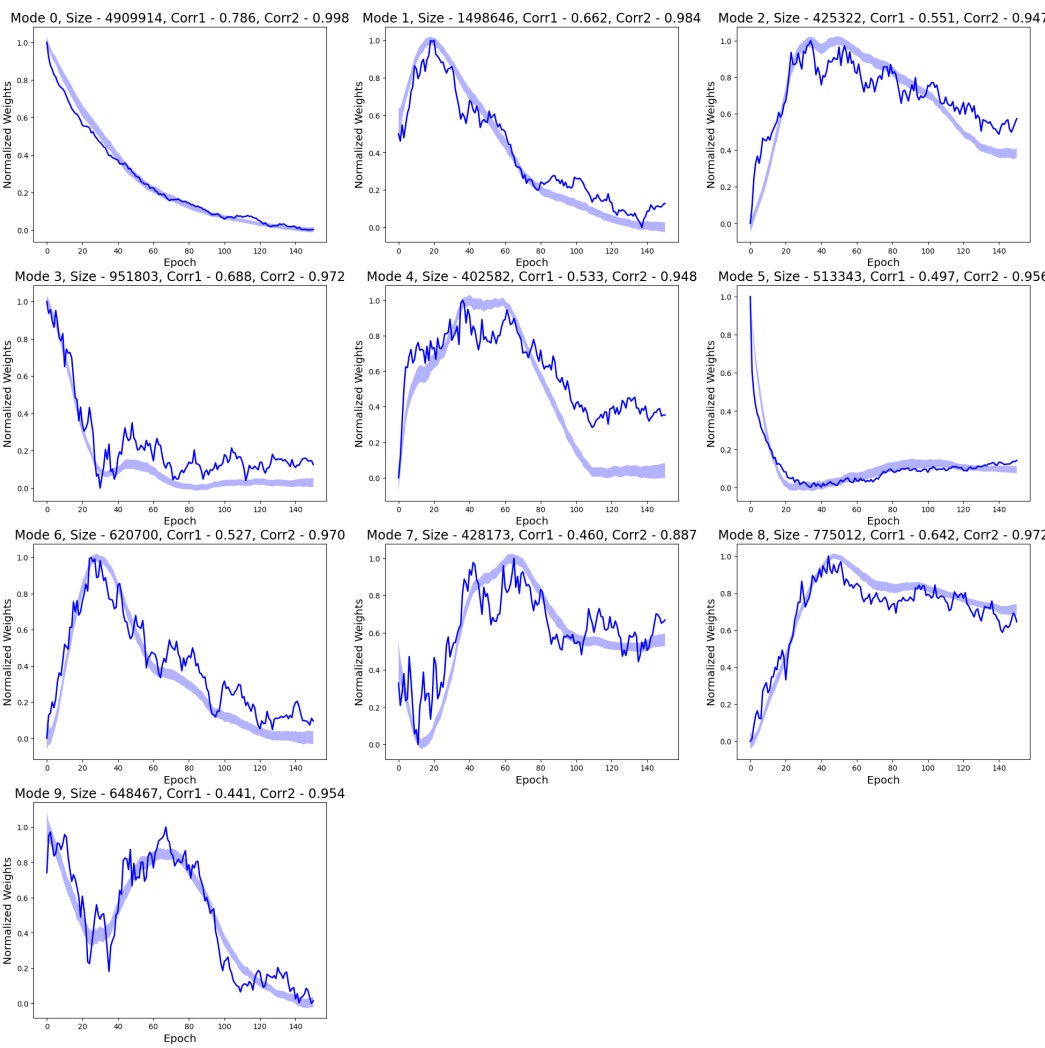

Figure 6: For each mode, weight trajectories are portrayed as mean and variance of the normalized trajectories (shaded area) alongside the reference trajectory (solid line). Each subplot represents a mode characterized by its parameter count (size), inter-mode correlation (corr1), and correlation of the mean trajectory with the reference trajectory (corr2). Formally, corr1 for the cluster $m_{th}$ cluster $C_m$ is $\frac{1}{\binom{N_m}{2}} \sum_{i \in C_m} \sum_{j \in C_m, j>i} \text{corr}(w_i, w_j)$, where $N_m$ is the size of the cluster, and corr2 is $corr\left(\frac{1}{N_m}\sum_{i \in C_m} w_i, w_m\right)$. Notably, corr2 consistently indicates a strong alignment of the mean trajectory with the reference trajectory, demonstrating its efficacy in capturing the general dynamics, even upon less-than-perfect values in corr1. Additionally, larger modes exhibit higher corr2 as well as corr1 values, suggesting more consistent behavior within the collective dynamics. The mode-average dynamics display varied patterns, including sharp decreases and subsequent stabilization, reflecting the learning process's adaptation phases. This diversity is effectively encapsulated by the reference trajectories, affirming our dynamics modeling approach despite the variation in correlation.

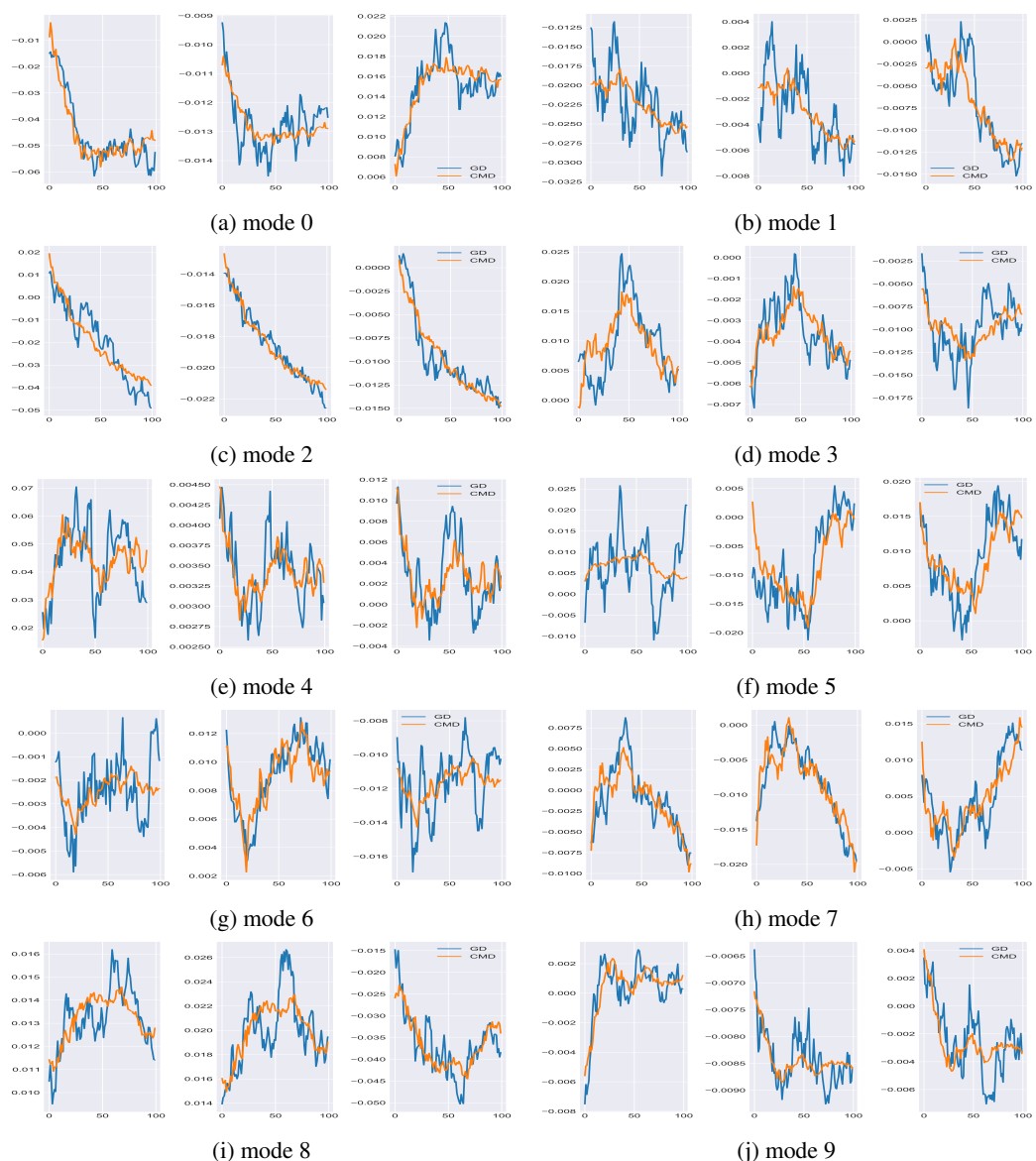

Figure 7: **CMD Weight Trajectory Reconstruction**. 3 randomly sampled weights from each mode, and their CMD reconstruction. CIFAR10 image classification, ResNet18. The CMD modeling provides more stable, less oscillatory solutions.

In Fig. 1 we compare CMD weight trajectory approximation to DMD weight trajectory approximation. Here we offer additional examples of CMD weight trajectory approximations. CMD is performed with $M = 10$ modes and 3 weights are randomly sampled from each mode for visualization. Manual observation is indispensable, therefore in Fig. 7 we show CMD smoothens the output, implying a regularization effect that may boost performance similarly to smoothing methods like EMA and SWA (Tarvainen & Valpola, 2017; Izmailov et al., 2018). Note however that EMA and SWA do not offer modelling (which is the core interest in our work).

## A.2 REFERENCE WEIGHT AND MODE WEIGHTS RELATIONS

In Fig. 8, statistical metrics validate CMD's weight trajectory modeling. The reference weight of a mode, together with the mean and variance of 10,000 randomly sampled weights per mode are presented. Different approaches are used - Post-hoc CMD (top row), Naive reference weight training

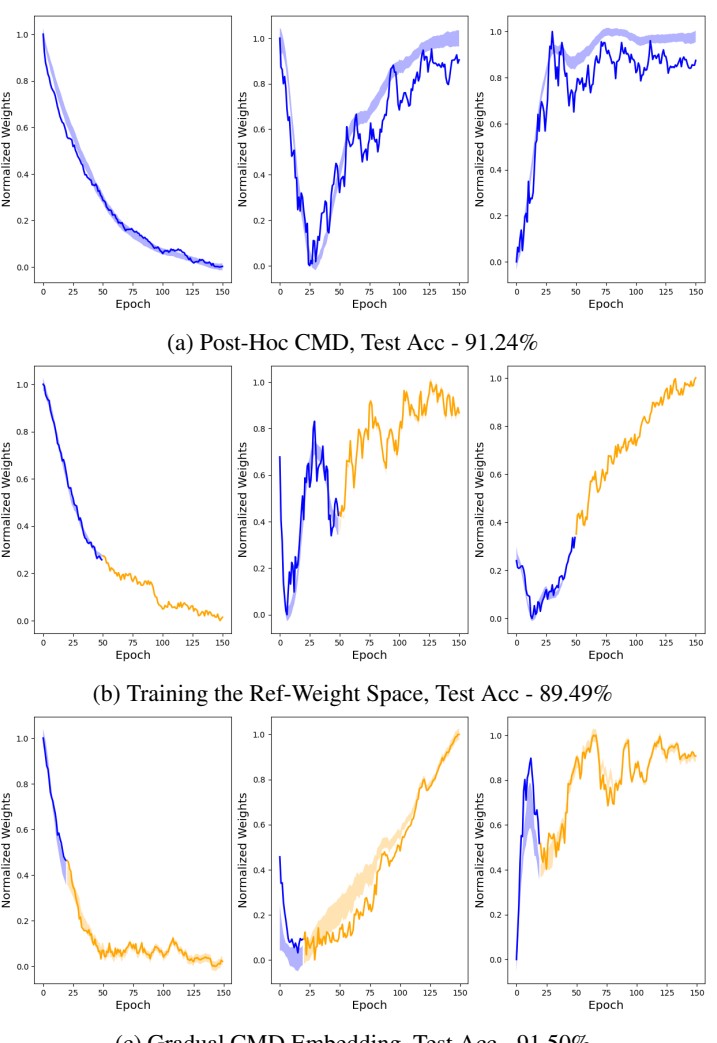

(a) Post-Hoc CMD, Test Acc - 91.24%

(b) Training the Ref-Weight Space, Test Acc - 89.49%

(c) Gradual CMD Embedding, Test Acc - 91.50%

Figure 8: **Reference Weight Trajectories with the Mean and Variance of Weights per Mode**. Top: post-hoc CMD; Middle: naive CMD embedding. Only reference weights are trained post warm-up phase; Bottom: Gradual CMD Embedding. Test Accuracy of the final model is available in the caption of each row. Blue color represents SGD trained weights and orange represents CMD embedded weights. The gradual embedding case mitigates high performance with a converging mean and decreasing variance.

(middle row) and gradual CMD embedding (bottom row). In each row 3 modes are presented. In the post-hoc CMD case the mean of the weights in the mode do not follow the exact dynamic of the reference weight, it is somewhat similar and more smooth. In addition, the variance increases throughout the epochs. In the naive reference weight training case all of the weights in the mode are directly related to the reference weight and change only according to the reference weight, post warm-up (orange). Therefore, there is no variance and the mean of the sampled weights is exactly equal to the reference weight. However, in this case the final test accuracy is not comparable to the other cases (more details about the accuracy are available in Fig. 3). The warm-up phase in this case was 50 epochs. In the gradual CMD embedding case the weights are gradually linked to the reference weights which results with a converging mean and decreasing variance. The high performance is maintained, in terms of test accuracy. The warm-up phase is 20 epochs.

Note: In order to display weights with different ranges of values in the same graph we stretch the values to the range of $[0, 1]$ - $\tilde{w}_i = \frac{w_i - min(w_i)}{max(w_i) - min(w_i)}$.

### A.3 Correlation Matrix Examples

Our main observation is that many network parameters, although non-smooth in their evolution, are highly correlated and can be grouped into "Modes" characterized by their alignment with a common evolutionary profile. In Fig. 1 we displayed two examples of a clustered correlation matrix. Bellow we present two additional examples. The first example is of SimpleNet (see Fig. 10b) trained on a binary classification task. In this case the parameters are divided into 3 modes. The main mode contains the majority of the network parameters. The second example is of a vision transformer, ViT-b-16, fine tuned on CIFAR10. The model parameters are divided into 30 modes. The modes contain a decreasing number of parameters, and some redundancy is visible. Redundancy in the number of modes is implied when all of the weights from one mode have high correlation to all of weights in a different mode (mode 0, mode 2, mode 4, etc. in this case, and mode 1, mode 3, etc.).

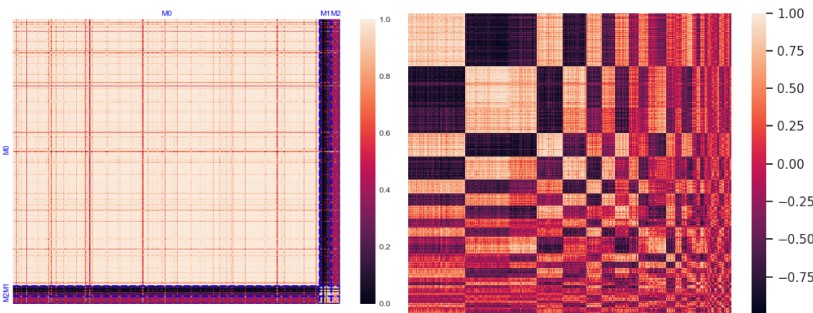

Figure 9: **Clustered Correlation Matrix**. Left - SimpleNet trained on MNIST, $M = 3$ modes. Right - ViT-b-16 fine tuned on CIFAR10, $M = 30$ modes. A single main mode is visible for SimpleNet, while for ViT-b-16 it appears there is a certain redundancy in the high number of modes.

## B  Post-Hoc CMD Experiments

**Image Classification**   First we consider the CIFAR10 (He et al., 2016a) classification problem, using a simple CNN architecture, used in Manojlović et al. (2020), we refer to it as *SimpleNet* (Fig. 10a). Cross-Entropy Loss and augmented data (horizontal flip, random crop, etc.) are used for training. The CNN was trained for 100 epochs, using Adam optimizer with momentum, and initial learning rate of $\eta = 1 \times 10^{-3}$. The CMD Analysis was applied on the full 100 checkpoints of this training. In this case we obtained 12 modes for the post-hoc CMD. The results from the CMD and the original SGD training are presented in Fig. 10c. We clearly see the model follows well the SGD dynamic, where both train and test accuracy are even slightly improved. In addition, the distribution of the modes through the entire network is presented in Fig. 10b. Each mode contains weights from each layer of the network. Therefor, trying to simplify the mode decomposition to full layers of weights is not expected to work as well. After validating our method on a simple scenario we tested CMD on larger architectures, such as ResNet18 (He et al., 2016a), see Fig. 12.

In recent years, after becoming a key tool in NLP (Devlin et al., 2018) transformers are making a large impact in the field of computer vision. Following Dosovitskiy et al. (2020), we apply our method on a pre-trained vision transformer. Our model was applied on the fine tuning process of a ViT-b-16 (Dosovitskiy et al., 2020; Gugger et al., 2021), pre-trained on the JFT-300M dataset (Sun et al., 2017), on CIFAR10 with 15% validation/training split. We used Adam optimizer with a starting learning rate of $5 \times 10^{-5}$ with linear scheduling. The network contains 86 million parameters. Negative Log Likelihood Loss was used. One can see in Fig. 2 that our method models the dynamics well and the induced regularization of CMD yields a stable, non-oscillatory evolution.

**Image Segmentation using PSPNet**   Our method generalizes well to other vision tasks. We present CMD modeling on the semantic segmentation task for PASCAL VOC 2012 dataset (Everingham & Winn, 2012). For this task we used PSPNet Architecture (Zhao et al., 2017), ($13 \times 10^6$ trainable parameters). The model was trained for 100 epochs, using SGD optimizer with momentum of 0.9, and weight decay of $1 \times 10^{-4}$. In addition, we used "poly" learning policy with initial

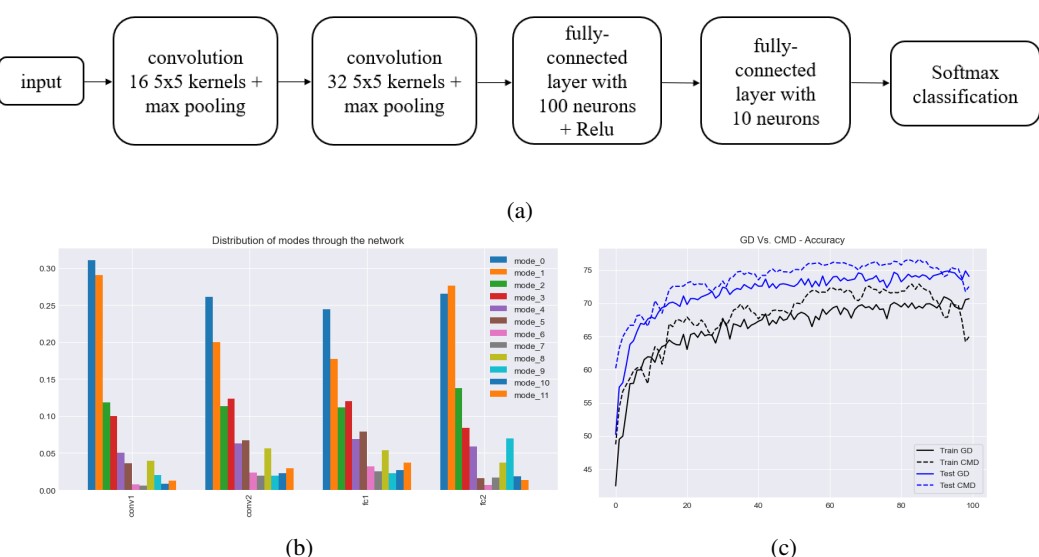

(a)

(b)                 (c)

Figure 10: **SimpleNet on CIFAR10**. (a) architecture (b) CMD Modes distribution through the Network. (c) CMD modeling performance

learning rate of $\eta = 1 \times 10^{-2}$. Post-hoc CMD was executed using $M = 10$ modes. Pixel accuracy and mIoU results from the CMD modeling and the original SGD training are presented in Fig. 11. Max CMD performance on the validation set: Pixel accuracy: 88.8% with mIoU of 61.6%. Max SGD performance on the validation set: Pixel accuracy: 88.1% with mIoU of 58.9%. We observe CMD follows SGD well in the training, until it reaches the overfitting regime. For the validation set, CMD is more stable and surpasses SGD for both quality criteria.

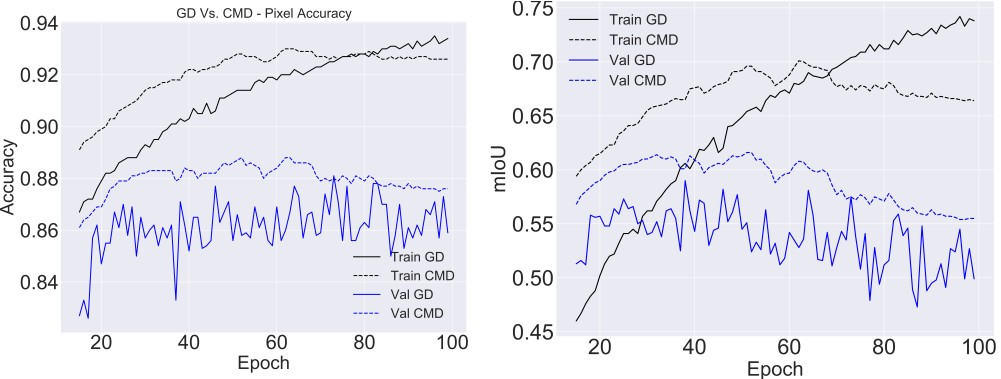

Figure 11: **Post-hoc CMD on PSPNet**. PASCAL VOC 2012 (Everingham & Winn, 2012) segmentation results, PSPNet Architecture. Original SGD training vs. CMD modeling with $M = 10$. modes. Left: pixel accuracy. Right: mIoU. CMD follows SGD well in the training, until it reaches the overfitting regime. For the validation set, CMD is more stable and surpasses SGD for both quality criteria.

**Image synthesis using StarGan-v2** We applied our CMD modeling on image synthesis task using StarGan-V2 (Choi et al., 2020). This framework consists of four modules, each one contains millions of parameters. It was modeled successfully by only a few modes, as demonstrated in Fig. 15.

**Ablation Study** Our method introduces additional hyper-parameters such as the number of modes $(M)$ and the number of sampled weights to compute the correlation matrix $(K)$. We performed

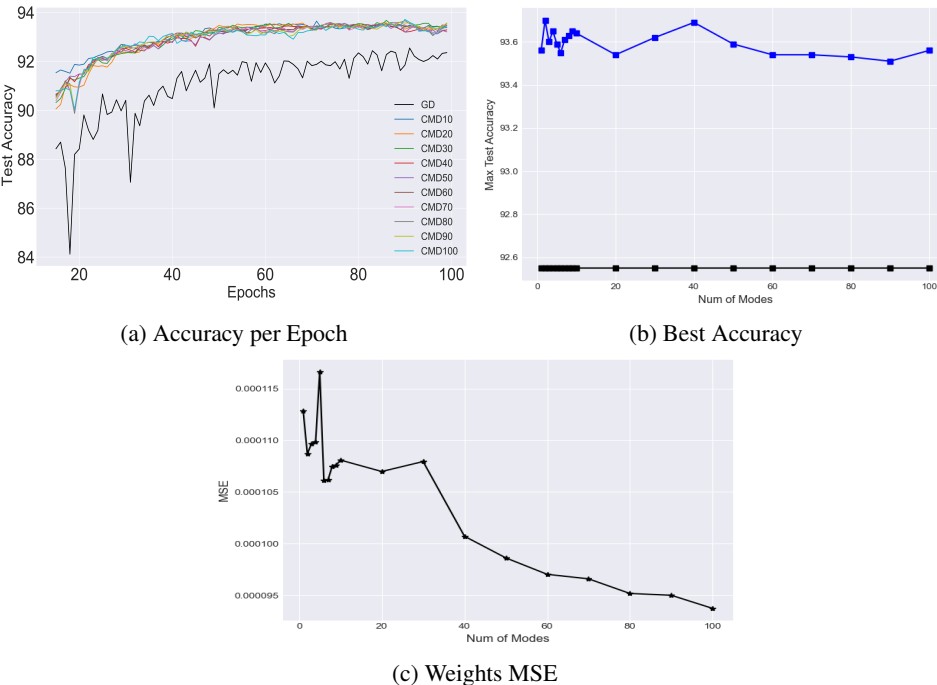

(a) Accuracy per Epoch                    (b) Best Accuracy

(c) Weights MSE

Figure 12: **ResNet18 CIFAR10 Results for Different Number of Modes** (a) Test Accuracy per epoch. (b) Maximal Test Accuracy of CMD model per number of modes used. (c) MSE of Weights reconstruction per number of modes used for CMD. While the MSE decreases as the number of modes increases, this is not reflected in performance, which is relatively constant.

several robustness experiments, validating our proposed method and examining the sensitivity of the CMD algorithm to these factors.

1. **Robustness to number of modes.** Fig. 12 presents the test accuracy results of 10 different CMD models, with different number of modes for the same SGD training. Surprisingly, the test accuracy is close to invariant to the number of modes. The best test accuracy per mode is presented in Fig. 12b. We reach the highest score for only two modes. It appears ResNet18 is especially well suited for our proposed model. Weights MSE does in fact reduce as more modes are generated (Fig. 12c), but this does not effect the model performance. It is important to note that the weights MSE is very low even for 2 modes.

2. **Robustness to different random subsets**. Here We examine empirically our assumption that in the K sampled weights we obtain all the essential modes of the network. In Fig. 13 we show the results of 10 CMD experiments executed on the same SGD training (ResNet18, CIFAR10), where in each experiment a different random subset of the weights is sampled. We fixed the number of modes to $M = 10$ and the subset size to $K = 1000$. The mean value of the test-accuracy at each epoch is shown, as well as the maximal and minimal values (over 10 trials, vertical bars). As can be expected, there are some changes in the CMD results. However, it is highly robust and outperforms the original GD training. In addition, we conducted several experiments, examining different values of $K$, the values were $K = 50, 125, 250, 500, 1000, 2000$. The number of modes was fixed ($M = 10$). For any fixed $K$, 5 experiments were carried out, see Fig. 13. We observe that the algorithm is robust to this meta-parameter as well. The results tend to be with somewhat less variance for larger $K$, but generally we see the sampling size for the initial clustering phase can be very small, compared to the size of the net $N$.

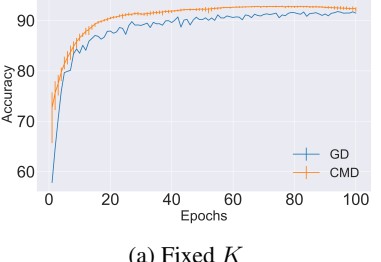
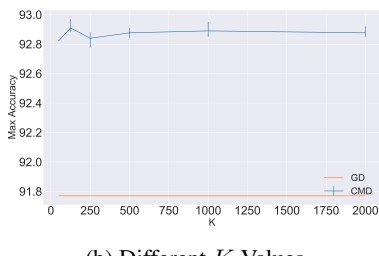

(a) Fixed $K$                     (b) Different $K$ Values

Figure 13: **ResNet18 CIFAR10 Results for Different Sampled Weights**. (a) Mean Test Accuracy per epoch for different experiments with fixed $K = 1000$. (b) Mean Test Accuracy per epoch for different experiments with different $K$ values. $M = 10$ in all the experiments. Despite different weights leading to slightly different models (a), CMD consistently outperforms SGD. Different sizes of $K$ yield very similar results (b).

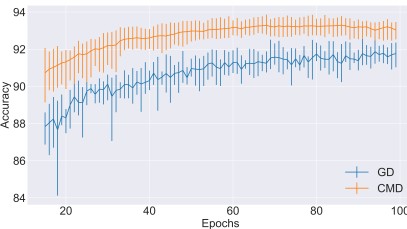

Figure 14: **ResNet18 CIFAR10 Results for Different Random Initializations**. Mean test accuracy (and variance) results for 10 different experiments is presented for SGD and CMD. The variance of the CMD results is similar to the variance of the SGD results, with a smoother mean.

3. **Robustness to random initialization** - In Fig. 14 we show that the effects of random initialization of the weights is marginal. The results of 10 training experiments are shown (ResNet18, CIFAR10). In each experiment a different random initialization of the weights is used. The same weights are sampled and the number of modes is fixed ($M = 10, K = 1000$). The variance of the CMD results is similar to the variance of the SGD results, with a smoother mean.

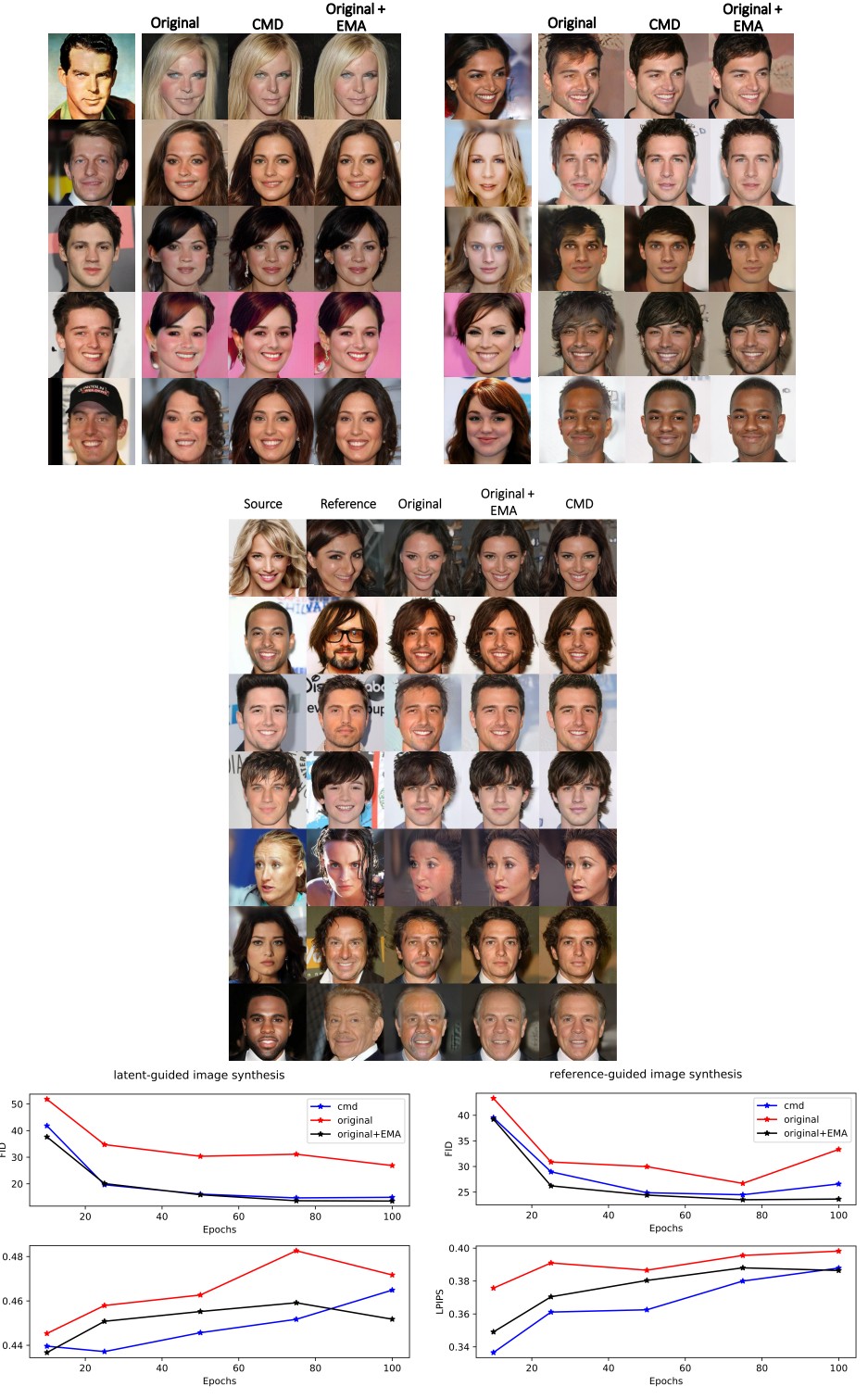

Figure 15: **StarGAN-v2 Style Transfer**. Post-hoc CMD modeling of the dynamics of StarGAN-v2 on Celeb-A-HQ dataset Karras et al. (2017). It can be seen that the CMD regularization effect is similar to that of the EMA in this style-transfer framework, both qualitatively, and in the image generation measures LPIPS and FID - which measure discrepancy from real Celeb-A-HQ images. CMD is performed for both reference-image and latent-noise guided generation.

# C    METHOD DETAILS

## C.1    ALGORITHMS

In Sec. 3.2 we introduce the CMD algorithm, consisting of three steps. Steps 1 and 2 describe clustering the model weights to modes. The clustering we use is a correlation based clustering with linear complexity and it is described in Algorithm 1. The full CMD algorithm is described, including the third step, in Algorithm 2.

---

**Algorithm 1** Correlation-based clustering with linear complexity

---

**Input:**
- $W \in \mathbb{R}^{N \times T}$ - Matrix of all weights.
- $K$ - Number of sampled weights.
- $M$ / $t$ - Number of wanted modes / desired in-cluster distance threshold.

**Output:**
- $\{C_m\}_{m=1}^M$ - Clusters of the network parameters.

**procedure**
1. Initialize $C_m = \{\}, m = 1..M$.
2. Sample a subset of $K$ random weights and compute their correlations.
3. Cluster this set to $M$ modes based on correlation values, update $\{C_m\}$ accordingly.
4. Choose profiles $w_m = \arg\max_{w_j \in C_m} \sum_{w_i \in C_m} |\text{corr}(w_i, w_j)|$.
   **for** $i$ in $(N - K)$ weights **do**
       $m^* = \arg\max_m |\text{corr}(w_i, w_{r,m})|$
       $C_{m^*} \leftarrow C_{m^*} \cup \{w_i\}$
   **end for**
   **return** $\{C_m\}_{m=1}^M$
**end procedure**

---

**Algorithm 2** CMD - Correlation Mode Decomposition

---

**Input:**
- $W \in \mathbb{R}^{N \times T}$ - Matrix of all weights.
- $K$ - Number of sampled weights.
- $M$ / $t$ - Number of wanted modes / desired in-cluster distance threshold.

**Output:**
- $\mathbf{W}^{Recon} \in \mathbb{R}^{N \times T}$ - CMD reconstruction of network parameters.

**procedure** CMD
    Perform correlation clustering to $M$ modes using Alg. 1. ($M$ is fixed if given, or deduced according to the $t$ input). Get mode mapping $\{C_m\}_{m=1}^M$.
    **for** $m \leftarrow 1$ to $M$ **do**
        Follow Eq. (6) to compute $A, B$, the affine parameters.
        Follow Eq. (7) to compute $\mathbf{W}_m^{Recon}$ per mode, the CMD reconstructed weights.
    **end for**
    **return** $\mathbf{W}^{Recon}$
**end procedure**

---

Algorithm 2 can return the CMD parameters ($A, B$ vectors, $w_m$ weights, etc.) instead of the reconstructed weights $\mathbf{W}^{Recon}$, if wanted. The Online CMD algorithm is presented in Algorithm 3.

## C.2    SAMPLING DETAILS

We would like to make sure that every layer is represented in the subset that is randomly sampled at the first step of our method (correlation and clustering). The choice of reference weights depends on

---

**Algorithm 3** Online CMD - Iterative AB Update Algorithm

**Inputs:**

- F - Number of warm-up phase epochs.

- E - Number of total epochs.

- Any other input needed for the SGD training (learning rate, momentum, etc.) or CMD algorithm (CMD sampled weights number - $K$, CMD number of modes - $M$, etc.)

**Outputs:**

- Regular SGD final model

- CMD final model

**Procedure**

1. Start SGD training and save model checkpoint in each epoch.

2. When $epoch = F$ perform the full CMD algorithm. All CMD parameters are saved (A, B, Related mode vectors and reference weights time profiles). No checkpoints (additional or previous) are needed.

3. For $epoch$ from $F + 1$ to $E$: update A,B vectors according to Eq. (9), using the current model weights. Update the reference weights time profiles.

4. After last epoch calculate final CMD model weights according to Eq. (7).

---

this set. We therefore allocate a budget of $\frac{K}{2 \cdot |layers|}$ for each layer and randomly sample it uniformly (using a total budget of $\frac{K}{2}$ parameters). The other $\frac{K}{2}$ parameters are sampled uniformly throughout the network parmeter space. When $\frac{K}{2} <= |layers|$ we simply sample all of the $K$ parameters uniformly. In the context of image classification, we observed a reduction in test accuracy variance when using this procedure (compared to naively sampling with uniform distribution accross all weights).

### C.3 CLUSTERING METHOD

To cluster the correlation map into modes we employed a full linkage algorithm - the Farthest Point hierarchical clustering method. After creating a linkage graph we broke it down into groups either by thresholding the distance between vertices ($\epsilon$), or by specifying a number for modes ($M$). Note - In Step 1, we associate $K$ weights; in Step 2, only the remaining $N - K$ weights are considered, reducing $C_2$ from $\mathbb{R}^N$ to $\mathbb{R}^{(N-K) \times M}$.

The Clustering method could be substituted by any clustering algorithm such as the simple k-means and others. We do not recommend methods such as PCA to avoid assumptions, such as orthogonality, which may not apply to weight trajectory space.

### C.4 ITERATIVE STEP 3 DERIVATION

Given $\tilde{A}_{m,t-1}$, we can re-express Eq. (6) as

$$\tilde{A}_m = \mathbf{W}_m \tilde{w}_m^T (\tilde{w}_m \tilde{w}_m^T)^{-1} \tag{17}$$
$$= \left[ \mathbf{W}_{m,t-1} \tilde{w}_{m,t-1}^T + \mathbf{W}_m(t) \tilde{w}_m(t) \right] (\tilde{w}_m \tilde{w}_m^T)^{-1},$$

On the other hand, by Eq. (6) at time $t - 1$ we have $\mathbf{W}_{m,t-1} \tilde{w}_{m,t-1}^T = \tilde{A}_{m,t-1} (\tilde{w}_{m,t-1} \tilde{w}_{m,t-1}^T)$. Plugging this we indeed get Eq. (9), which is:

$$\tilde{A}_m(t) = \left( \tilde{A}_{m,t-1} (\tilde{w}_{m,t-1} \tilde{w}_{m,t-1}^T) + \mathbf{W}_m(t) \tilde{w}_m^T(t) \right) (\tilde{w}_{m,t} \tilde{w}_{m,t}^T)^{-1}, \tag{18}$$

### C.5 POST-HOC VS ONLINE CMD COMPARISON

Post-Hoc CMD is the basic approach, while Online CMD is a highly efficient version of it. While it is often the case that efficiency comes at the expense of performance - here it is not the case,

as validated by our experiments, see Table. 4. The comparison is done on ResNet18 trained on CIFAR10 with different number of modes. The post-hoc experiments are done at the end of the full training (150 epochs). The online approach is performed with a short warm-up phase (20 epochs) for steps 1 and 2 in the online CMD algorithm (see Sec. 3.3). The iterative updates are then performed until the end of training (from epoch #21 to the last epoch - epoch #150). Each experiment is performed 5 times, on the same SGD training process. The same weights are sampled for both approaches in each experiment. In Fig. 16 we present an experiment comparing online CMD runs with different warm-up phases. We examine short warm-up phases (20,30,40 epochs) and longer warm-up phases (80, 100, 120 epochs). We also present the regular SGD training case and the post-hoc CMD case. All CMD runs, online CMD with different warm-up phases and post-hoc CMD, achieve similar results. CMD is performed with 10 modes in all the experiments in this figure, using the same training process.

Table 4: **Comparing Online CMD vs the basic Post-hoc CMD**. Accuracy performance is comparable, where the online version has slightly higher variance. The number in the model column represents the number of modes.

| Model | Test Accuracy |
|---|---|
| CMD1 (online) | $92.06\% \pm 0.05$ |
| CMD1 (post-hoc) | $92.06\% \pm 0.02$ |
| CMD2 (online) | $92.07\% \pm 0.07$ |
| CMD2 (post-hoc) | $92.01\% \pm 0.04$ |
| CMD5 (online) | $91.94\% \pm 0.12$ |
| CMD5 (post-hoc) | $91.96\% \pm 0.05$ |
| CMD10 (online) | $91.98\% \pm 0.17$ |
| CMD10 (post-hoc) | $92.05\% \pm 0.25$ |

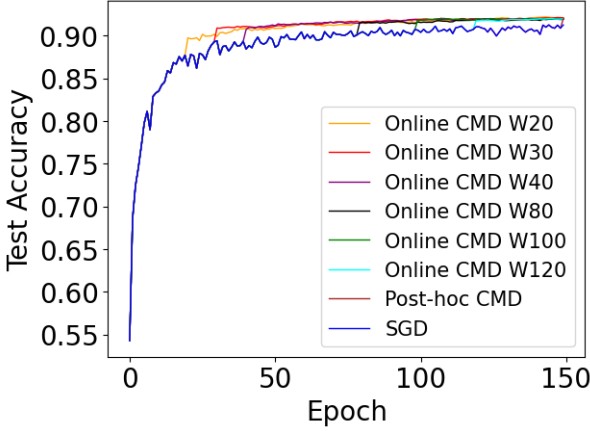

Figure 16: **Online CMD with different warm-up phases**. Test accuracy for online CMD with different warm-up phases. The value after W in the legend descries the length of the warm-up phase. SGD and post-hoc CMD results are presented as well.

## C.6   ONLINE CMD FLOPS COMPUTATION

Here we show rigorously that the Online CMD algorithm computation can be completely neglected compared to the SGD which it accompanies. Upon the online CMD Eq. (9), Eq. (7) are performed on each epoch that occurs after the $F$ warm-up epochs (note that Eq. (7) is performed only in order to validate the model and is not necessary for the training process itself. However we will take the worst case when calculating the computational overhead.) We consider $t_2 = t_1 + 1$, which we also use in practice. Let us calculate the number of FLOPS required for each epoch in this case. Reminder: an $n \times p$ matrix multiplied by a $p \times m$ matrix takes $nm(2p - 1)$ FLOPS.

Let us start with Eq. (9), which is given by:

$$\tilde{A}_{m,t2} = (\tilde{A}_{m,t1}(\tilde{w}_{m,t_1}\tilde{w}_{m,t_1}^T) + W_{m,t_1:t_2}\tilde{w}_{m,t_1:t_2}^T)(\tilde{w}_{m,t_2}\tilde{w}_{m,t_2}^T)^{-1}. \tag{19}$$

We begin with the complexity of the first term:

$$\tilde{A}_{m,t1}(\tilde{w}_{m,t_1}\tilde{w}_{m,t_1}^T). \tag{20}$$

Computational complexity: First multiplication of a $N_m \times 2$ by a $2 \times 2$ is $6N_m$ FLOPS. The second multiplication (inside the brackets), may be updated iteratively on its own, similarly to Eq. (8), with 8 FLOPS, i.e. we treat it as $O(1)$. Thus we have $6N_m + O(1)$ FLOPS.

The last term:

$$(\tilde{w}_{m,t_2}\tilde{w}_{m,t_2}^T)^{-1}. \tag{21}$$

The multipication is similarly $O(1)$ (via another iterative update), and the inverse of a $2 \times 2$ is $O(1)$ as well.

The middle term:

$$W_{m,t_1:t_2}\tilde{w}_{m,t_1:t_2}^T. \tag{22}$$

Multiplying a $N_m \times t_2 - t_1$ by a $t_2 - t_1 \times 2$, considering $t_2 - t_1 = 1$ yields $2N_m$ FLOPS.

The first addition takes $2N_m$ FLOPS, and the remaining multipication by the inverse matrix is a $N_m \times 2$ miltiplied by the inverse which is a $2 \times 2$ resulting with $6N_m$ FLOPS.

In total summing all FLOPS and summing over the modes, where the number of modes is much smaller than $N$, we have

$$14N + O(1) \text{ FLOPS}. \tag{23}$$

Now let us move to the easier Eq. (7). Here we have one element-wise multipication and one elementwise addition to the weights vector, i.e. $2N$ FLOPS.

Thus in total we have $16N + O(1)$ FLOPS, that can be done on the CPU (do not require GPU). This is negligible considering typical actual FLOPS, for instance - ResNet18 has $\frac{70.07 \cdot 10^9}{1.82 \cdot 10^6}N = 38,500N$ FLOPS just for a forward pass, according to the model zoo provided here.

## C.7 ONLINE CMD MEMORY CONSUMPTION

The final memory consumption of the algorithm, for a random number of modes, after the $F$ warm-up epochs, is approximately 2.25 times the memory of a single model. The CMD parameters, and there relative sizes are detailed below.

1. Vectors A and B each have a size equal to the size of a full model.
2. $\tilde{M}$ vector which stores the related mode of each weight has the size of $\approx 0.25$ times the model. This vector has the same length as A and B, but the values stored have a smaller range - positive integers that represent the mode number. This vector can be compressed even more if necessary.
3. The full time profile for each reference weight. A $2 \times 2$ matrix can be stored for each reference weight instead, as explained in Step 3 in Sec. 3.3. Both cases have negligible size compared to the full model size.

It is well established (see for instance Izmailov et al. (2018)) that the space occupied by storing the model is insignificant during the training process compared to the sustained gradients. Therefor, after the $F$ warm-up epochs this online method has negligible memory consumption.

## C.8 COEFFICIENT STABILIZATION

Analyzing the changes in the affine parameters $\{a_i, b_i\}_{i=1}^N$ throughout training reveals variations among them, with some experiencing more significant changes than others. Additionally, as depicted in Fig. 17, the portion of parameters that remain relatively unchanged increases as the training procedure progresses. This motivates us to gradually embed the CMD model into the SGD training process.

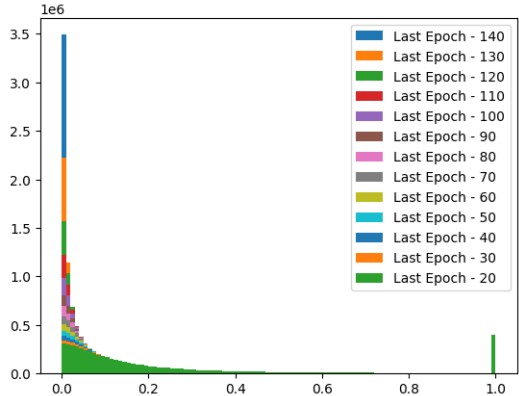

Figure 17: **Relative Change in the Affine Parameters**. Histogram of the relative change in the $\{b_i\}_{i=1}^N$ parameters calculated in different epochs. The relative change is compared w.r.t. $\{b_i\}_{i=1}^N$ at the end of the process, at epoch 150. As we can see, there are weights that do not change their value at all between epochs 20 and 150. Additionally, the portion of B parameters that do not change, or change a little, increases throughout the training procedure.

## D  RESULTS SECTION EXPERIMENTAL DETAILS

**Models**  In order to generalize our method and verify that it works on various models and architectures we examined different ResNet models and networks with different architectures. We strive to check models with different properties in order to examine a diverse set of examples. In the ResNet family of models we tested ResNet18, Preactivation ResNet-164 (He et al., 2016b) and Wide ResNet28-10 (Zagoruyko & Komodakis, 2016). PreRes164 is a very deep network composed of more than 800 layers. The number of parameters in this model is 2.7 million, less than in ResNet18 (11.2 million). In Wide ResNet28-10 the feature channels per layer are multiplied by 10 and the depth, the number of residual blocks, is 28. This model has 36.5 million parameters, more than three times the number of parameters in ResNet18. A different architecture, which is very basic, is LeNet-5 (LeCun et al., 1998). This model, which was presented more than 20 years ago, has a very basic architecture, composed of several convolutional layers. We also tested in GoogleNet (Szegedy et al., 2015), another model which is not relatively new. In addition to these well known models we also demonstrate the CMD methods presented in this work on a vision transformers architecture named ViT-b-16 introduced in Dosovitskiy et al. (2020). Transformers have become a fundamental tool in NLP (Devlin et al., 2018) and many computer vision tasks (Dosovitskiy et al., 2020). The Vit-b-16 model we used was pre-trained on ImageNet1k (Russakovsky et al., 2015). The code we used for each model is linked to the following model names. ResNet18, Wide ResNet28-10, Preactivation-ResNet-164, LeNet-5, GoogleNet, ViT-b-16.

Table 5: **SGD Training Implementation Details**. In ViT-b-16 a cosine scheduler is used, see more details in text. For generalization different optimizers are used.

| Model | Training Epochs | Optimizer | Learning Rate |
|---|---|---|---|
| ResNet18 | 150 | SGD | 0.05 |
| WideRes | 150 | SGD | 0.1 |
| PreRes164 | 150 | SGD | 0.1 |
| LeNet-5 | 150 | Adam | 0.001 |
| GoogleNet | 120 | Adam | 0.001 |
| ViT-b-16 | 90 | SGD | 0.03* |

**Implementation Details**  Each experiment is performed 5 times and the mean test accuracy and standard deviation are presented in the results tables (Tables 1, 2). The training parameters (number of epochs, optimizer, learning rate) and the CMD parameters (warm-up phase, number of modes ($M$), number of sampled weights ($K$)) for each model are presented in Tables 5, 6 respectively.

Table 6: **CMD Implementation Details**. Models with relatively high number of layers (PreRes164, GoogleNet) require more warm-up epochs and more sampled weights for stable performance. For generalization different number of modes are used.

| Model | Warm-up Epochs | Modes | Sampled Weights |
|---|---|---|---|
| ResNet18 | 20 | 10 | 1000 |
| WideRes | 20 | 5 | 1000 |
| PreRes164 | 40 | 2 | 10000 |
| LeNet-5 | 20 | 10 | 1000 |
| GoogleNet | 30 | 7 | 5000 |
| ViT-b-16 | 20 | 10 | 1000 |

In all the experiments the weight decay was 0 and when SGD is used momentum is 0.9. In the ViT-b-16 training we used a cosine scheduler starting with a short (5 epochs) warm-up phase where the learning rate linearly increases from 0 to 0.03. Then a cosine decreasing scheduler is applied for the remaining 85 epochs. The code for this scheduler is from here. The training parameters used are training parameters used in other works, specifically in the linked projects from above. Additionally, we wanted to show CMD works with any optimizer, not specifically SGD, so we used Adam in several cases. Both SGD and ADAM optimizers are implemented using the Pytorch library (Paszke et al., 2019). The CMD warm-up phase was enlarged for relatively deep models, with many layers. The number of modes used in each experiment was not optimized. Rather, various number of modes from 2 to 10 modes were chosen for generalization purposes. P-BFGS code is provided by the authors of the original paper here. In Fig. 18 single run comparisons between the regular SGD test accuracy and the Online CMD test accuracy are presented. Each plot describes a single run, on a specific model (see figure caption), following the experimental details described above. The test accuracy of the Online CMD method is much more smooth compared to the regular SGD training result.

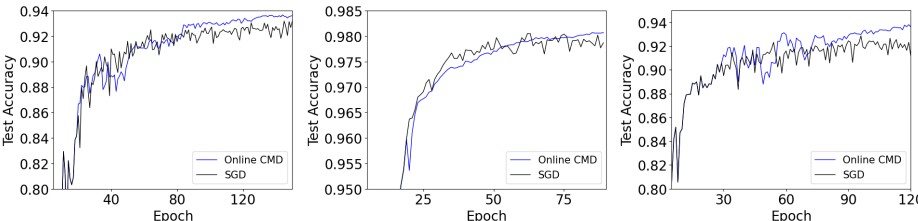

Figure 18: **SGD vs Online CMD Comparison**. Results of a single run are presented for each model. The Online CMD is superior to the regular SGD in all cases. Left: WideRes28-10, Middle: ViT-b-16, Right: GoogleNet.

For Table 2 we used the same learning rate in all experiments ($LR = 0.05$), when momentum was applied the value was - 0.9 and the learning rate scheduler (when used) is the Cosine scheduler with no warm-up epochs. All the training sessions are 150 epochs long, with a warm-up phase of 20 epochs for CMD. EMA is performed with a smoothing factor $\beta = 0.9$ and SWA is performed on the last 25% of epochs. CMD was performed with $M = 10$ modes.

In order to demonstrate CMD on a different dataset we present a small experiment, performed on CIFAR100 (Krizhevsky et al., 2009), in Table 7. The implementation details of this experiment on ResNet18 are - 150 epochs, learning rate - 0.05 and momentum - 0.9. CMD is performed with 10 modes, the number of sampled weights (K) is 1000 and 20 warm-up epochs were used. For Embedded CMD we used L - 10 and P - 10. The experiment is repeated 5 times, the mean and standard deviation of the results are presented.

In Table 2 results of CMD are compared to EMA and SWA, which all achieve similar results. To further analyze the relations between these methods we experimented with preforming one on top of the other (Table 9. Using ResNet18 on CIFAR10, we reconstructed the full model weight values per epoch for EMA and CMD (post-hoc). Then, we performed SWA, EMA and CMD (post-hoc) on each set of weights - SGD weights, EMA weights and CMD weights. The implementation details

Table 7: **CMD on CIFAR100.** Both CMD methods (online and embedded) improve the baseline SGD training.

| Model | Baseline SGD | Online CMD | Embedded CMD |
|---|---|---|---|
| ResNet18 | $66.81\% \pm 0.62$ | $\underline{68.74\% \pm 0.44}$ | $\mathbf{69.27}\% \pm 0.57$ |

Table 8: **CMD on TinyImageNet.**

| Model | Baseline SGD | Post-hoc CMD |
|---|---|---|
| ResNet18 | $26.01\% \pm 0.31$ | $28.4\% \pm 0.72$ |

of this experiment are - 150 epochs, learning rate - 0.05 and momentum - 0.9. CMD (post-hoc) is performed with 10 modes and the number of sampled weights (K) is 1000. The results indicate that similar regularization introduced by EMA and SWA is attained by CMD as well.

Table 9: **CMD relations with EMA and SWA:** The test accuracy at the end of the baseline SGD training is $91.24\%$. Using only one of the methods (SWA, EMA or CMD) leads to similar results as combining two methods together. This experiment is performed once, therefore there is no mean or standard deviation.

| Weights/Method | SWA | EMA | CMD |
|---|---|---|---|
| SGD | $92.1\%$ | $92.07\%$ | $92.06\%$ |
| EMA | $92.04$ | - | $92.07\%$ |
| CMD | $92.09\%$ | $92.07\%$ | - |

# E EMBEDDING CMD DETAILS

## E.1 ALGORITHM

The full gradual CMD embedding and weight freeing algorithm is presented in Algorithm 4.

---

**Algorithm 4** Gradual CMD Embedding Algorithm

---

**Hyper-parameters:**
    $F$ (Number of warm-up epochs);
    $L$ (Number of epochs between each embedding epoch);
    $P$ (Percentage of embedded coefficients in each embedding epoch);
    Any inputs required for the CMD algorithm or training process (learning rate, etc.)

**Procedure**
    $W_{1:F} \leftarrow$ weight trajectories of $F$ regular SGD epochs.
    $A, B,$ related modes $\leftarrow$ Post-hoc CMD on $W_{1:F}$.
    $\mathcal{I} \leftarrow \emptyset$                              ▷ index set of embedded weights
    **for** $t > F$ **do**
        Regular SGD step
        Iterative update of $A$, $B$ via Eq. (9)
        $w_i \leftarrow \begin{cases} \text{SGD update} & \text{if } i \notin \mathcal{I} \\ a_i w_r(t) + b_i & \text{if } i \in \mathcal{I} \end{cases}$ ▷ in accordance with Eq. (11), where if $i \in \mathcal{I}$ then $a_i, b_i$ are fixed in time
        **if** $(t - F)\%L == 0$ **then**
            $Diff \leftarrow \sqrt{(A_{\text{old}} - A)^2 + (B_{\text{old}} - B)^2}$     ▷ excluding reference weights and embedded weights
            $\{A, B\}_{\text{old}} \leftarrow \{A, B\}$
            $\mathcal{I} \leftarrow \mathcal{I} \bigcup \{\text{indices of the bottom } P \text{ percentile of } Diff\}$
        **end if**
    **end for**

---

### E.2 DIFFERENT EMBEDDING SCHEDULING

We examine different values for P, as well as several approaches to define $P$, in our experiments. $P$ can be defined as the percentage from all the weights in the model, this is noted as the basic case. In this case using a constant value $P$ will lead to the same number of weights embedded in each embedding epoch. A different way to define $P$ is the percentage of the not embedded weights. This relative percentage approach is noted as *relative P*. Applying a constant value $P$ will lead to a reducing number of weights embedded on each embedding epoch, as the number of not embedded weights reduces. Reducing the value of $P$ is another approach. One can decrease the value of $P$ linearly, exponentially, etc. In addition a scheduled decrease can be used as well. In Sec. 5, in the federated learning experiments, a combined approach involving both a constant $P$ and an exponentially decreasing $P$ is used (see Appendix F.1). We demonstrate these different approaches bellow in Fig. 19. We examined different $P$ values $\{10, 20\}$, for the basic case and the *relative P* case. In addition we used a pre defined scheduled decreasing P, starting from $P = 20$. We used ResNet18 trained and tested on CIFAR10. The CMD warm-up phase is 20 epochs and $L = 10$ in all cases. The mean test accuracy of 5 different training cycles is presented, with the final accuracy and standard deviation available in the legend. Overall, from the five different options presented, embedding the A,B vectors iteratively, with $P = 20\%$ and using the relative percentage approach yields the best results. However, results are very similar for most cases.

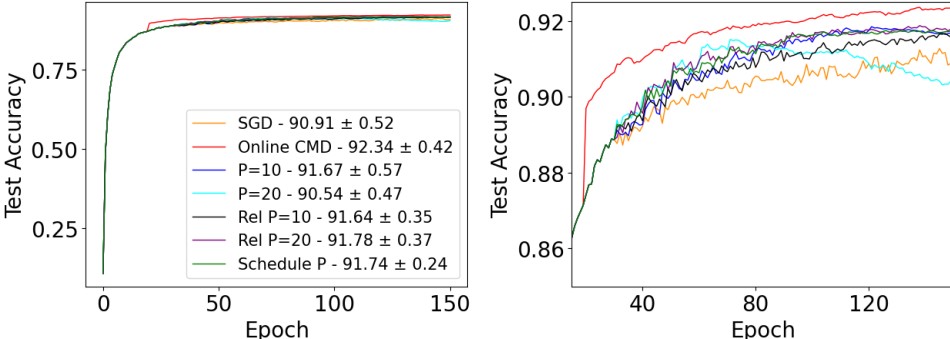

Figure 19: **Gradual Weight Embedding with Different P Scheduling**. Results are improved compared to SGD training, while there is some drop in performance compared to the regular CMD. The results are very close, most options achieving nearly the same results. The only drop in performance occurs in the basic approach with $p = 20$, where after only 70 epochs (five embedding epochs) all of the weights are embedded.

### E.3 EMBEDDED CMD VS ONLINE CMD

We compare the Online CMD algorithm (Algorithm 3) to the gradual CMD embedding algorithm (Algorithm 4) on different models, see Table 10. In these experiments the implementation details are the same as the experiments in Table 1. The Embedding algorithm parameters are fixed for all experiments - $L = 10, P = 10$, using the basic constant $P$ embedding approach. The gradually embedded CMD method introduces weight freezing which enhances efficiency. Therefore, a limited drop in performance is reasonable. For some models the chosen hyper-parameters achieve better results than the full online CMD method.

Table 10: **Comparing Online CMD vs Embedded CMD**. In some cases the Embedded CMD outperforms the Online method, in other cases the online method with no embedding is superior.

| Model | Baseline SGD | Online CMD | Embedded CMD |
|---|---|---|---|
| ResNet18 | $90.91\% \pm 0.52$ | $\mathbf{92.34\%} \pm 0.42$ | $91.80\% \pm 0.30$ |
| WideRes | $92.70\% \pm 0.48$ | $\mathbf{93.21\%} \pm 0.50$ | $92.36\% \pm 0.77$ |
| PreRes164 | $90.64\% \pm 0.98$ | $91.34\% \pm 1.20$ | $\mathbf{91.71\%} \pm 0.58$ |
| GoogleNet | $92.05\% \pm 0.48$ | $93.07\% \pm 0.49$ | $\mathbf{93.36\%} \pm 0.14$ |

# F   FEDERATED LEARNING DETAILS

## F.1   FEDERATED LEARNING EXPERIMENT DETAILS

We closely followed the experimental setting provided in the original paper: 50 Clients with disjoint datasets of non-IID class distribution that is drawn independently for each client from a Dirichlet distribution with $\alpha = 1$. 10% of the clients are sampled for synchronization , i.e. $|C[t_s]| = 0.1|\mathcal{C}| \forall t_s$. 1,500 rounds. Each synchronization round, each client completes 10 epochs. The APF threshold is set to $\mathcal{T} = 0.05$ , and whenever the number of embedded coefficients crosses $80\%$, perform $thresh \leftarrow \frac{thresh}{2}$. To conform with this scheduling, we also perform for our method $P \leftarrow \frac{P}{2}$ under the same circumstances. For CMD we set $P = 5\%$. The 'aggressiveness' parameter schedule of A-APF is set as $\min\left(\frac{round}{2000}, 0.5\right)$.

## F.2   APF CODE

Weight freezing can be combined with other strategies to enhance efficiency in Federated Learning. GlueFL He et al. (2023) stands out as a framework that adeptly merges weight freezing with client sampling, ensuring optimized communication overhead, especially crucial in cross-device settings like mobile or IoT devices. They have open-sorce code, including comparison to Chen et al. (2021) in here. We used their project for our APF implementation.

## F.3   CALCULATION OF THE NEGLIGIBLE ADDED COMMUNICATION

Let us show that $\frac{\frac{2N}{E}}{\hat{N}^{\text{not-embedded}}(|C|+|\mathcal{C}|)} \sim 10^{-4}$. In our setup of ResNet18, we have $N \sim 10^7$. Following Chen et al. (2021) we have $|\mathcal{C}| = 50$, $|C| = 5$. As shown in Fig. 5 these result with convergence for $E \sim 10^3$. Thus we have $\frac{\frac{2N}{E}}{\hat{N}^{\text{not-embedded}}(|C|+|\mathcal{C}|)} \sim \frac{\frac{2N}{E}}{\hat{N}^{\text{not-embedded}}|\mathcal{C}|} = \frac{2}{E|\mathcal{C}|}\frac{N}{\hat{N}^{\text{not-embedded}}} = \frac{2}{5\cdot10^4}\frac{N}{\hat{N}^{\text{not-embedded}}}$. In our experiments $\frac{\hat{N}^{\text{not-embedded}}}{N} \approx 0.5$, resulting with $\frac{1}{5\cdot10^4} \approx 10^{-4}$.

## F.4   POTENTIAL OF GENERALIZED DYNAMICS MODELING IN FEDERATED LEARNING

Our work with CMD in FL has demonstrated promising preliminary results. A key contribution here is leveraging dynamics modeling for efficient communication in distributed learning, and is not restricted to CMD. This section discusses the potential of extending the framework to include various dynamics modeling approaches, emphasizing the value it may serve in FL.

The core of this generalized approach remains the use of dimensionality reduction. By communicating low-dimensional representations of model parameters between clients and the central server, the volume of transmitted data can be significantly reduced. Alongside the transmission of compressed data, it is crucial for client models to effectively reconstruct the original dynamics from the reduced information. Maintaining local copies of necessary data and mappings enables clients to accurately translate the low-dimensional representations back to the full parameter space.

With dimensionality reduction at its core, we believe this approach may allow for a more interpretable understanding of the model's behavior. For instance, by focusing on a reduced set of parameters, it becomes easier to trace and understand how these parameters influence the learning outcome. This is particularly beneficial in FL environments, where understanding the contribution of individual clients to the collective model is often complex and opaque. The ability to map back and forth between the full and reduced parameter spaces not only ensures efficient communication but also provides a clearer insight into the dynamics of the learning process. This transparency has the potential to aid in diagnosing the learning process for various issues.

As the field of dynamics modeling continues to evolve, new dynamics modeling approaches will emerge. A generalized framework, not limited to CMD, can seamlessly integrate these techniques, ensuring continued effectiveness and relevance in a rapidly evolving domain. Moreover, a natural extension is to generalize the framework to accommodate a range of dynamics modeling techniques. This generalization will utilize the most suitable dynamics model for specific tasks.

In summary, the generalization of the CMD framework to include a variety of dynamics modeling approaches presents an intriguing prospect. We believe such an approach can remain relevant and

adapt for future advancements in FL as well as dynamics modeling. The goal is to enhance the adaptability, efficiency, and scalability of FL, providing a foundation for further research and development in this area. The addition of explainability to this framework not only aids in effective communication but may also enhance our understanding of distributed learning processes. Thus we believe this could be a valuable tool for future advancements in the field.

# G  CMD LANDSCAPE VISUALIZATION: EXTENSION

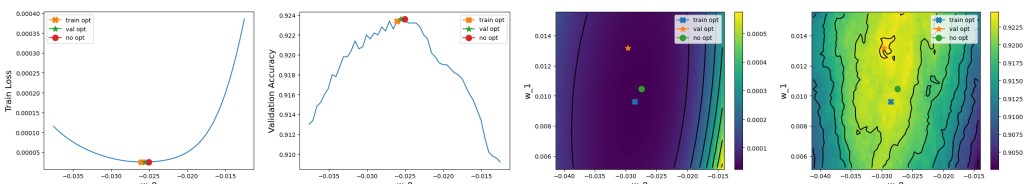

Figure 20: **Loss and Accuracy Landscapes for Post-hoc CMD**. Left - CMD with a single mode, 1D curves; Right - CMD with two mode, 2D landscapes. The optimal models from each dataset, and the original model are marked on each landscape and described in the legend.

A different feature available due to the reduced number of parameters of the model is a visualization feature. Shifting the reference weight values of CMD with a single mode will lead to a 1D loss curve. In the 2 modes case a 2D loss landscape is created. Starting from post-hoc CMD with a single mode, in Fig. 20 we present the 1D curves and 2D landscapes representing the train loss and validation accuracy metrics for different values of the reference weights. The values of the reference weights determine the value of all the other reconstructed weights in the model, as shown in Eq. (7). Changing the reference weights values changes the values of all the other weights as well, creating a new model, easily calculated. An array of values are tested in the area of the original reference weights values - $[w_0 - |\frac{w_0}{2}|, w_0 + |\frac{w_0}{2}|]$ with $\frac{w_0}{50}$ increments (1D case), or - $[w_0 - |\frac{w_0}{2}|, w_0 + |\frac{w_0}{2}|] \times [w_1 - |\frac{w_1}{2}|, w_1 + |\frac{w_1}{2}|]$ with $\frac{w_0}{50}, \frac{w_1}{50}$ increments (2D case).

The main visual difference between the landscapes of the training loss and the validation accuracy in both cases is that the training set loss is smooth and the validation set accuracy is not. For the training set case there is a clear global minimum. For the validation set case there are many local maximum points, around the global maximum. We will note that the training set accuracy behaves similarly to the training set loss function and that the validation set loss behaves similarly to the validation set accuracy function. The smooth landscape in the training set case indicates that the CMD model is much more stable on the training set, compared to the validation set. Generated from the training data, this is an expected feature of the CMD model.

Fine tuning the model on a specific class (or classes) of data could also be done via this visualization method. An example is presented in Fig. 21, where each 2D landscape presents the validation accuracy landscape for images of a specific class. For each class different values of $(w_0, w_1)$ optimize the validation set accuracy. The 2D landscapes produced per class show different behaviour from the full validation set accuracy landscapes presented earlier in this section. These 2D landscapes are divided into regions with the same exact performance. These different landscapes show how each model calculated with different reference weights values affects each class differently. The caption of each sub-figure contains the name of the class and the test accuracy of the optimized model in brackets. The original test accuracy, before any optimization, is 91.99%. Some classes lead to better optimized models (bird, frog, etc.) and others lead to a lower overall test accuracy (airplane, deer, etc.).

A different approach is embedding a subset of the CMD model parameters into the model trained by SGD. The embedded parameters are dependant on the reference weights and the other parameters have fixed values. In Fig. 22 we illustrate an example (ResNet18 on CIFAR10) where 10% of all the model weights are embedded, 30% are embedded and 100% of the model weights are embedded. As expected, when more weights are embedded, and linked to the reference weights, a more significant change is visible in the 2D loss landscape.

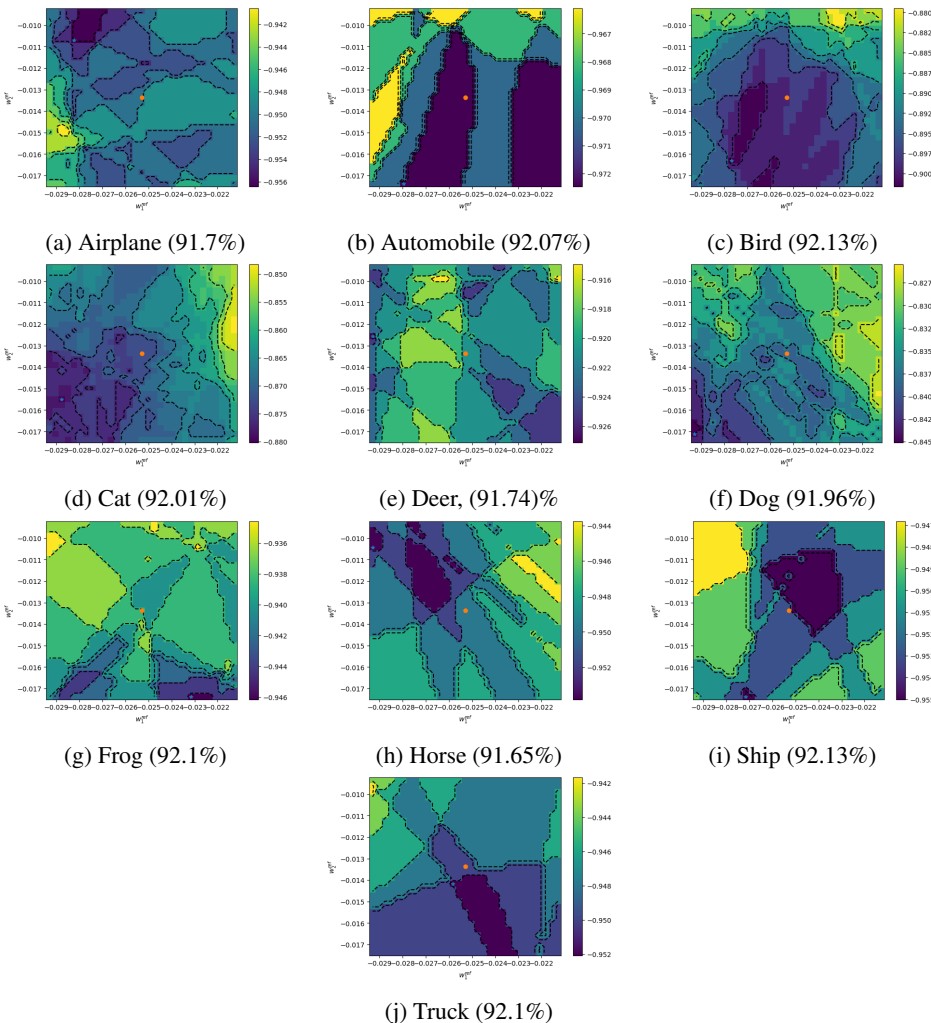

Figure 21: **2D Validation Accuracy per Class Landscapes for CMD with Two Modes**. The caption of each sub-figure contains the name of the class and the test accuracy of the optimized model in brackets. The original test accuracy, before any optimization, is 91.99%. Some classes lead to better optimized models (bird, frog, etc.) and others lead to a lower overall test accuracy (airplane, deer, etc.).

To conclude this section, using the reduced number of parameters for visualization intents shows interesting options for different subsets of the data or different subsets of the model parameters. The visualization also leads us to the conclusion that there are many similar models, that achieve similar results, in the area of the original CMD calculated model. By changing the values of the reference weights we can easily calculate a new model, with different weight values, that achieves similar results.

# H  DMD ANALYSIS FOR NEURAL NETWORKS

## H.1  DYNAMIC MODE DECOMPOSITION

*DMD* is one of the common and robust tools for system analysis. It is widely applied in dynamical systems in applications ranging from fluid dynamics through finance to control systems (Kutz et al., 2016). DMD reveals the main modes of the dynamics in various nonlinear cases and yields constructive results in various applications (Dietrich et al., 2020). Thus, it is natural to apply DMD-

Figure 22: **Loss Landscapes for Post-hoc CMD with Different Portions of the Embedded Model**. Left - 10% of the CMD model weights are embedded; Middle - 30% of the CMD model weights are embedded; Bottom - 100% of the CMD model weights are embedded; A stable behaviour is maintained when different portions of the CMD model are embedded in the SGD trained model. When a larger percentage of the CMD model weights are embedded, the changes are more visible in the landscape. The maintained minima as well as smoothness throughout the subplots suggests that incorporating partial CMD modeling in training, as in embedded CMD, is viable

type algorithms to model the process of neural network training in order to analyze the emerging modes and to reduce its dimensions. For instance, in Naiman & Azencot (2021) it was proposed to use DMD as a method to analyze already trained sequential networks such as Recurrent Neural Networks (RNNs) to asses the quality of their trained hypothesis operator in terms of stability.

DMD is originally formulated as an approximation of Koopman decomposition (Mezić, 2005), where it is argued that any Hamiltonian system has measurements which behave linearly under the governing equations (Koopman, 1931). These measurements, referred to as Koopman eigenfunctions, evolve exponentially in time. This fact theoretically justifies the use of DMD, where it can be considered as an exponential data fitting algorithm (Askham & Kutz, 2018). The approximation of Koopman eigenfunctions by DMD can generally not be obtained for systems with one or more of the following characteristics (Cohen & Gilboa (2023)):

1. Finite time support.
2. Non-smooth dynamics - not differential with respect to time.
3. Non-exponential decaying profiles.

Below we demonstrate a simple example of applying DMD on a single-layer linear network, showing that already in this very simple case we reach the limits stated above when using augmentation.

### H.2 DMD IN NEURAL NETWORKS

The training process of neural networks is highly involved. Some common practices, such as augmentation, induce exceptional nonlinear behaviors, which are very hard to model. Here we examine the capacity of DMD to characterize a gradient descent process with augmentation. We illustrate this through a simple Toy example of linear regression.

**Toy example – linear regression with augmentation.** Let us formulate a linear regression problem as follows,

$$\mathcal{L}(x, w; y) = \frac{1}{2}||y - wx||_F^2 \tag{24}$$

where $||\cdot||_F$ denotes the Frobenius norm, $x \in \mathbb{R}^{m \times n}$, $y \in \mathbb{R}^{d \times n}$ and $w \in \mathbb{R}^{d \times m}$. In order to optimize for $w$, the gradient descent process, initialized with some random weights $w_0$, is

$$w^{k+1} = w^k + \eta^k \left(y - w^k x\right) x^T, \tag{25}$$

where the superscript $k$ denotes epoch number $k$, $T$ denotes transpose and $\eta^k$ is a (possibly adaptive) learning rate. We note that introducing an adaptive learning rate by itself already makes the dynamic nonlinear, as $\eta$ denotes a time interval. Let us further assume an augmentation process is performed, such that at each epoch we use different sets of $x$ and corresponding $y$ matrices, that is,

$$w^{k+1} = w^k + \eta^k (y^k - w^k x^k)(x^k)^T. \tag{26}$$

Then, the smoothness in Eq. (25) becomes non-smooth in the augmented process, Eq. (26). Actually, we obtain a highly nonlinear dynamic, where at each epoch the operator driving the discrete flow is different. We generally cannot obtain Koopman eigenfunctions and cannot expect DMD to perform well. Our experiments show that this is indeed what happens in practice. In the modeling of simple networks, DMD can model the dynamics of classical gradient descent but fails as soon as augmentation is introduced. Fig. 23 presents an experiment of CIFAR10 classification using a simple CNN (Fig. 10a) with standard augmentations, usually performed on this dataset (horizontal flips and random cropping). In this experiment in order to reconstruct the dynamics, we perform DMD with different dimensionality reduction rates ($r = 10, 50, 90$) using Koopman node operator, as described in Dogra & Redman (2020). The results presented in Fig. 23 show that when the dimensionality reduction is high ($r = 10$) DMD fails to reconstruct the network's dynamics. However, when the dimensionality reduction is mild ($r = 90$) DMD reconstruction is also unstable and oscillatory. An example of DMD performed on a more complex network (ResNet18), with no data augmentation, is also available in Fig. 23.

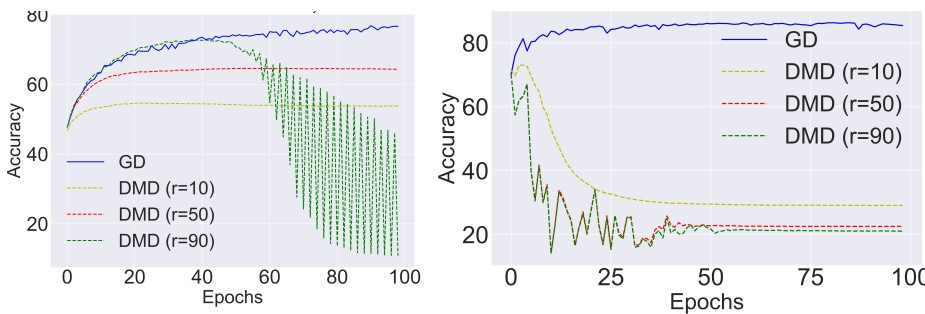

Figure 23: **DMD Experiments**. CIFAR10 classification results, Original Gradient Descent (GD) training Vs. DMD modeling. Top: SimpleNet architecture, with augmentations. Bottom: more complex model - ResNet18, no augmentations. 3 values of dimensionality reduction parameter r per experiment. DMD fails to achieve comparable results in all cases.

### H.3 DMD AND CMD

The common ground between DMD and CMD is the intention to approximate a complex dynamic with a simpler one. In both cases, the original dynamic is decomposed into time profiles and spatial structures. The main difference is that DMD uses for its modes pre-known eigenfunctions, in a linear approach, which is limited to exponential profile representation. CMD is nonlinear, using a single representative with improved adaptability. DMD employs Eq. (1), while we employ the special case of Eq. (2).

Several key differences exist - first, DMD is computed by finding a linear operator that approximates the dynamic with a least-squares problem. The operator is later decomposed into spatial structures (modes) and eigenvalues yielding exponential profiles in time. CMD, conversely, is not based on pre-known temporal functions, but rather assumes correlation between parameters and uses time profiles present in the dynamic itself. They are not computed using an eigen-decomposition but rather by clustering a correlation matrix and finding a single representative profile in each cluster. Secondly, there is no operator approximation in CMD, thus it does not have, at least directly, the option to extrapolate the dynamic by continuing applying the operator to the end of the training data. Thirdly, decomposition in CMD is "transposed" in relation to DMD, in the sense that the CMD "modes" are groups with similar temporal behavior (a cluster of the correlation matrix), while in DMD each mode is a spatial function that has to be multiplied by an exponent to form its contribution to the total. To match this logic, the CMD modes should have been a vector including all coefficient vectors (a, b) from all clusters. But then, the whole vector does not have the same temporal behavior, as it is divided to clusters. For CMD with a single mode - the coefficient vector is similar to the DMD notion of a spatial mode.

