# OpenReview forum: "Enhancing Neural Training via a Correlated Dynamics Model"
_ICLR.cc/2024/Conference — ICLR 2024 poster_

### Official Review · Reviewer_JHvF · 2023-10-29

**Soundness:** 4 excellent
**Presentation:** 4 excellent
**Contribution:** 4 excellent
**Rating:** 10
**Confidence:** 4

**Summary:**

The authors make a novel hypothesis about the way in which deep neural network weights dynamically evolve during training: "weights can be clustered into a very few highly correlated groups" (Hypothesis 1). The authors illustrate support for this hypothesis by examining the training of several different types networks on several different tasks, using post-hoc correlation based clustering. The success they found on these examples motivated them to develop an online method, which reduced computational demands and led to better accuracy than state-of-the-art low dimensional training methods. Lastly, the authors provided initial results on improving federated learning (and lowering computational costs) via their method.

**Strengths:**

1. This paper was well motivated and written. It was (for the most part) very easy to follow.

2. The hypothesis of highly correlated modes fits well within existing work (which the authors cite well), but is a highly novel discovery. This makes it impactful and interesting.

3. The results the authors achieved are impressive. They performed better than standard methods of training, and achieved state-of-the-art results among low-dimensional training methods. They tested on a variety of tasks (including federated learning, which - the authors say - makes it the first use of dynamical modeling to reduce communication overhead).

4. The Appendices are full of examples, pseudo-code, clarifying remarks, and extra details. This is a very packed paper.

**Weaknesses:**

I found no major weaknesses of this paper.

Here are a few points that whose clarification will enhance the quality of the paper:

1. The notation used in Sec. 3.3 (Online CMD) is a little difficult to follow. Additionally, because the use of online CMD to train DNNs had been foreshadowed in earlier parts of the paper, I was confused by the lack of details on how the individual modes were trained - which was answered in Sec. 3.4. Explicitly mentioning, in Sec. 3.3., that more details are coming on the actual training are coming in Sec. 3.4, would help remove this momentary confusion.

2. I did not understand the comment "Even though $\tilde{A}_m$ is re-evaluated at each time-step $t$, each $a_i, b_i \in \tilde{A}_m(t)$ are fixed scalars." (Sec. 3.3). Is this pre-empting the point made in Sec. 3.4 that, once embedded, a weight's $a_i, b_i$ are frozen?

3. I felt like Sec. 3.5 (CMD Accuracy and Loss Landscape Visualization) is too quickly presented. What is the main takeaway? That you can use CMD to visualize the landscape, or that CMD training does a good job of finding the optimal parameters? Maybe this would be better in the Appendix?

4. I think the work on federated learning is very interesting and a great application of the approach. That being said, I did feel like the explanation of the approach was a little rushed (possibly due to lack of space) and Eq. 16 was a bit confusing. Improving the discussion around Sec. 4 would be helpful (and could be aided by putting Sec. 3.5 in the Appendix to get more room).

Minor Comments:

1. The acronyms in Table 2 should be clarified (I assume Mom is momentum but what is SCH?).

2. Could you use DMD/Koopman methods to predict how the individual modes are going to evolve, thereby reducing the computational cost even more?

3. The comparison of DMD with CMD in Fig. 1 is insightful. Recent work has found that using Graph Neural Networks can lead to better approximations of the Koopman operator, specifically for the training of DNNs [1]. Do you think the correlated modes you discover might explain this improved success (e.g., the GNN is able to better learn the correlated modes)?

4. There are several places where the first quotation marks in a quote are backwards. This can be remedied by using `` instead of '' in Latex.

5. In Fig. 2, CMD is compared to GD. Should this be SGD?

[1] https://arxiv.org/abs/2305.09060

**Questions:**

All my questions are posed in the section above.

**Details Of Ethics Concerns:**

I have no ethical concerns.

---

> ### Author Response · Authors · 2023-11-15
>
> Hello,
>
> Thank you for your time and valuable comments, as well as points for discussion.
>
> **Weakness 1** - To improve the flow of the paper, we added the following opening paragraph to Sec. 3 (it includes the previous opening of subsection 3.2 and further details).
>
> "We present several approaches to apply our method, starting by discussing the underlying correlated dynamics (Sec. 3.1). Followingly we introduce the basic CMD version, called post-hoc CMD, which analyzes dynamics after they occur (Sec. 3.2). We then present the efficient online CMD (Sec. 3.3) and embedded CMD (Sec. 3.4) which embeds modeled dynamics into the training process. In the last subsection we demonstrate accuracy landscape visualization using CMD."
>
> **Weakness 2** - Below we provide a detailed explanation about this point. Since it is not very important, yet somewhat confusing - we decided to remove it from the manuscript to avoid the misunderstanding of future readers.
>
> Detailed explanation:
>
> The message here is that the coefficients are always used as static scalars for modeling. These coefficients, though subject to iterative updates during the training process as detailed in Section 3.3, are consistently employed as static scalars.
>
> **Scalar Nature of CMD Coefficients:** In all CMD modelings, the coefficients \(a_i, b_i\) are used as scalars throughout the process. Even when we model the dynamics at different epoch ranges (e.g., epochs 1-100 and 1-200), which results in different coefficient sets, each set is a set of scalar coefficients.
>
> **Iterative Updates in Online CMD:** In the online CMD approach, these coefficients are updated iteratively throughout the training process. However, it's crucial to understand that at each iteration, the coefficients \(a_i, b_i\) are re-evaluated and assigned new scalar values. They do not evolve into or represent time-dependent trajectories such as \(w_i(t)\). Each update is a discrete change, resulting in a new set of scalar values.
>
> **Potential Misconceptions:** The iterative update process in online CMD could suggest to readers a trajectory-like behavior for these coefficients, as they are updated through time (and trajectories are time-dependent). However, this is not the case. Despite the dynamic nature of the updates, the coefficients \(a_i, b_i\) do not transition into trajectories; they are updated as distinct, individual scalar values at each time step.
>
> For instance, if at time t, we would like to approximate the dynamics at time interval [t1, t2]:t2<t, then we would use the coefficients obtained at time t (the latest available time).
>
> **Weakness 3** – The visualization options available when using CMD introduced in Section 3.5 are quickly and succinctly presented.  The main takeaway is that CMD could be used for visualization of the loss/accuracy landscape, as another interesting application that CMD offers. Appendix G is an extended section presenting different visualization results for different scenarios. Although most of the information is available only in the appendix, we decided to shortly present this idea in the main paper, so that interested readers will know to look for the expansion in the Appendix.
>
> **Weakness 4** – We have added an explanation regarding Eq. (16) (and omitted details on the visualization section). Additionally, in Appendix F.3 we have added a new discussion that outlines the advantages of using dynamics modeling for efficient communication in Federated Learning, beyond the scope of CMD.  It highlights the potential for improved model interpretability and adaptability to future dynamics modeling techniques, enhancing efficiency, scalability, and understanding in Federated Learning. See also Appendices F.1,F.2 for technical details and communication overhead complexity of our FL experiments.

---

> ### Author Response · Authors · 2023-11-15
>
> **Minor Comments 1, 4, 5** – In Figure 2, 'GD' in the legends of the plots stands for SGD. A clarification has been added to the figure caption. Additionally, we added an explanation for the acronyms in Table 2 - 'Mom' stands for momentum and 'SCH' stands for leaning rate scheduler. All the backward quotation marks have been fixed as well. Thank you for pointing these issues out.
>
> **Minor Comment 2** - The use of DMD/Koopman methods in this context is interesting. If it works - it can open the possibility to extrapolate the reference trajectories forward in time, and in turn forward the full set of trajectories.
>
> **Minor Comment 3** - Thank you for bringing this research to our attention. We were not acquainted with this approach for accommodating highly nonlinear dynamics with DMD. In fact, it may alleviate concerns we raise in Appendix H.2 (DMD in Neural Networks). We added it to the Previous work section. The question raised is interesting. We think there might be a connection, we note there are still some differences between this work and our method. Mainly, in that paper the model dynamics are approximated using a set of  eigenfunctions (as in other DMD approaches) and in CMD the dynamics are approximated by reference dynamics from the original trajectories. However, We do think that the use of a GNN might assist learning eigenfunctions that are more aligned to the system (similarly to our method). We anticipate additional fruitful research in this direction.

---

> > ### Comment · Reviewer_JHvF · 2023-11-15
> >
> > I thank the authors for their detailed responses.
> >
> > As mentioned in the original review, I think this work is excellent and the changes that the authors made - which satisfy my (limited) concerns - further strengthen the quality of this work.
> >
> > I have no further comments.

---

### Official Review · Reviewer_yr3M · 2023-10-31

**Soundness:** 2 fair
**Presentation:** 1 poor
**Contribution:** 2 fair
**Rating:** 5
**Confidence:** 2

**Summary:**

This paper proposes a novel method (Correlation Mode Decomposition, CMD) to reduce the dimension of learning dynamics of neural networks, by leveraging correlations between learning dynamics. Using correlations between (the entire histories of) learning dynamics of each parameters, the proposed method first divides the set of parameters into several clusters (called modes),  identifies one representative parameter from each modes, and then represents the other parameters in each mode by scaling the representative and adding a bias. The paper also provides an online version of the dimensionality reduction (Online CMD) that can be used without memorizing the history of parameters during training, which still requires training of all parameters, and a parameter-efficient version (Embedded CMD) that enables us to reduce the number of trainable parameters gradually during training. The paper also propose to use the dimensionality reduction of learning dynamics for distributed learning. The paper empirically shows the superiority of CMD against standard training and the state-of-the-art methods of dimensionality reduction.

**Strengths:**

1. The procedure of CMD seems reasonable and also novel in dimensionality reduction of learning dynamics.
2. It may be also novel that their proposal to use the dimensionality reduction for distributed training, but less confident since I'm not an expert in this area.
3. Experimental results (Figure 3) shows a surprising result that Online/Embedded CMD outperforms full SGD on CIFAR-10, which seems somewhat contradictory because Online/Embedded CMD was designed to approximate the full SGD.

**Weaknesses:**

1. There are many unclear points in experimental results/figures.
    1. In each mode block in Figure 1 (Left & Middle), correlation between most of parameters tends to be less than 1.0, which does not satisfy the hypothesis behind the proposed method: `Any two time trajectories u, v that are perfectly correlated can be expressed as an affine transformation of each other`.
    2. The y-axis in Figure 1 (Right) is unclear.
    3. What are the different/common points between CMD and DMD? The paper highlights CMD vs DMD in Figure 1, but any description for DMD is not provided.
    4. Any theoretical evidence of Hypothesis 1 is not provided.
    5. `Figs. 1, 6 demonstrate that Eq. (2) accurately represents the dynamics using a notably small set of modes` in Section 3.1 is overclaimed due to W1-1 and W3.
    6. Accuracies plotted in Figure 2 (Left) seem inconsistent with the corresponding accuracy in Table 1. It is also weird that test accuracy of CMD is consistently higher than training accuracy of CMD in Figure 2.
    7. The author claimed that `we observed that the reference trajectories can be selected once during training, and do not require further updates`. However I could not find such results in the paper.
    8. In Table 1, it is weird that CMD significantly outperforms full training (GD) although CMD is designed to approximate GD. At least, CMD should be worse than GD in training accuracy if CMD behaved as a regularizer as the authors claimed. If CMD outperforms GD even in training accuracy, I'm concerning that there should be some leakage or bug in experiments.
2. Experiments are limited on a single toy dataset CIFAR-10. I'm concerning whether the proposed method still works well on more large-scale, difficult learning tasks.
3. Since CMD identifies only single representative for each cluster, the method only leverage proportional relationships between parameter dynamics, which may lead to very limited expression power of the reduction method, especially in more complex learning scenarios.

**Questions:**

See Weaknesses.

---

> ### Author Response · Authors · 2023-11-15
>
> Hello,
> Thank you for your time, comments and insights. Below, you will find our detailed response, supplemented by further experiments and references to the manuscript, addressing the weaknesses you have highlighted.
>
> **Strength 3 and Weakness 1.8** - The inherent boost in performance was initially surprising to us as well. See the Results section, under “CMD as a regularizer”, where we attribute this (beneficial) phenomenon to smoothing of the trajectories, i.e. reduction of parameter fluctuations - which are typical upon SGD. The boost in train and test accuracy upon such smoothing was shown and explained in previous work, see for instance [1], [2], which introduced Exponential Mean Averaging (EMA) and Stochastic Weight Averaging (SWA) respectively. We compare ourselves to [1], [2] in Table 2.
>
> **Weakness 1.1** - Eq. (2)'s approximation of mode dynamics varies in preciseness. Consequently, correlation varies as well, as indicated by Fig. 1 as you noted. This raises questions about Eq. (2)’s representation of the original dynamics upon less-than-perfect correlation. To answer these questions we added an experiment (Fig. 6 in Appendix A.1) that visualizes the original dynamics alongside their representation while displaying average mode correlation. It shows good approximation of the general dynamics even upon less-than-perfect-correlation.
>
> **Weakness 1.2** - These are the weight values, added the label “weight” to the y-axis, and the label “epoch” to the x-axis.
>
> **Weakness 1.3** - DMD is described in Appendix H. We added to Appendix H another subsection H.3, about the commonalities and differences of DMD and CMD. To summarize: DMD employs predetermined eigenfunctions in its linear method, restricted to representing exponential profiles. CMD, on the other hand, is nonlinear and utilizes a single representative with enhanced data-driven adaptability. DMD utilizes Eq. (1), whereas our approach adopts a specific instance of Eq. (2). Both are methods which try to capture low-dimensional representation of the dynamics.
>
> **Weakness 1.4** - We agree additional investigations are required to substantiate our findings with a corresponding theory. From a theoretical point of view - it is known that NNs exhibit large redundancies, hence low-dimensional representations should exist. While our correlation-based model is novel, it aligns with previous research, see for instance [3], which strengthens its utility. From a practical point of view - acting upon our hypothesis indeed results with reduced redundancy while exhibiting high performance.
>
> **Weakness 1.5** - Given the \approx sign in Eq. (2) it is indeed more fitting to replace "accurately represents” with “approximates” (regarding W1.1 and W3 see our response). Fixed.
>
> **Weakness 1.6** -
>
> **a.** Fig. 2 and Table 1 present distinct experiments, each fine-tuning a different pre-trained Vit-b-16 model: Fig. 2 uses post-hoc CMD with a model pre-trained on JFT-300M (dataset of 300M samples), while Table 1 employs online CMD with a model pre-trained on ImageNet1K (1.2M samples). We acknowledge that this should have been clearly stated, and revised the captions of Fig. 2 and Table 1 to include this distinction. We additionally added Fig. 18 in Appendix D, an experiment that aligns with the experiment setting of Table 1.
>
> **b.** The title should read “Baseline SGD” and not just “training” (as in “regular training”). Thank you for noticing, it is fixed now. As Table 1’s caption suggests, this whole table solely consists of test accuracy results. For further details on these experiments see Tables 5, 6 in the Appendix D.
>
> **Weakness 1.7** - In order to support our claim we added Fig. 16 to Appendix C.5. This figure illustrates online CMD's performance with varying warm-up durations, indicating that extended warm-up periods do not impact the outcomes. Table 4 in Appendix C.5 implicitly shows this claim as well: This table compares post-hoc CMD with online CMD. The former selects the reference trajectories based on the full training trajectories and the latter selects reference trajectories using a short 20-epoch warm-up period. The table demonstrates comparable performance between the two methods over various numbers of modes.
> To conclude, Both the added experiment in Fig. 16 and Table 4 in the Appendix support that there is no need to wait until the end of training before selecting the reference trajectories. References to Table 4 and Fig. 16 have been added in the main body of the paper to support the claim.

---

> ### Author Response · Authors · 2023-11-15
>
> **Weakness 2** - Regarding the concern about limited datasets and transferability to large-scale difficult tasks:
>
>
> **a. Data Scope:** Besides Cifar10, our research also incorporates CelebA-HQ for detailed image generation and PASCAL VOC 2012 for image segmentation, offering a variety of datasets.
>
> **b. Architectural Scale:** Our work encompasses a range of architectures. This includes smaller models like LeNet, mid-scale ones such as ResNets, and larger frameworks like WideReNet (36.5M parameters),  ViT-b-16 (86M parameters). We also experiment with the StarGAN-v2 (50M parameters, composing a complex of Generator, Style Encoder, a Latent-Noise Mapping networks), demonstrating our engagement with large-scale and challenging architectures.
>
> **c. Learning Tasks:** A concern about carry-over to other difficult learning tasks was raised. While our results section does focus on image classification (as is the case in previous work and benchmarks of this domain), we conduct further experiments, showing the effectiveness of CMD in image segmentation and image generation, particularly exploring the dynamics of GAN training. Moreover, we believe that we are the first to test such methods for image generation.
>
> **Weakness 3** - See our answer to **Weakness 2** regarding complex learning scenarios \ difficult tasks.
>
> Investigating a generalized version, as you suggest (as in Eq. (1)), is a proper direction for future research. We do remark that DMD, which uses a linear combination approach as you proposed, has limitations in expressiveness. This is detailed in our analysis in Appendix H.2, titled “DMD in Neural Networks,” where we discuss the constraints of DMD in representing complex learning dynamics.
>
> Though simple, our approach, which is data-driven and nonlinear, shows modeling capabilities of complex dynamics in various scenarios. This underscores the ongoing challenge of balancing expressiveness and simplicity in model reduction techniques.

---

> ### Comment · Reviewer_yr3M · 2023-11-21
>
> Thank you for your detailed reply and clarification. I still have concerns about W1-8 and W2.
>
> > The inherent boost in performance was initially surprising to us as well. See the Results section, under “CMD as a regularizer”, where we attribute this (beneficial) phenomenon to smoothing of the trajectories, i.e. reduction of parameter fluctuations - which are typical upon SGD. The boost in train and test accuracy upon such smoothing was shown and explained in previous work, see for instance [1], [2], which introduced Exponential Mean Averaging (EMA) and Stochastic Weight Averaging (SWA) respectively. We compare ourselves to [1], [2] in Table 2.
>
> I think this point should be clarified in more scientific way, not just by analogies, because it would be one of the most surprising part in this paper if correct. For example, if CMD behaved as a regularizer, training accuracy would be suppressed or similar to vanilla SGD, since CMD is designed to just approximates SGD. Are there such evidence for "CMD as a regularizer"? Also, do similar phenomena (i.e., both of boosted accuracy and accelerated training) occur with other datasets for image classification? Although the author claims they also used CelebA-HQ and PASCAL VOC in Appendix (with only 2 figures), they only conducted all experiments on CIFAR-10 other than them. Since CIFAR-10 is a simplified toy dataset and can yield misleading conclusions, the claim should be verified on other real-world datasets for image classification such as Caltech256, ImageNet, Oxford Flower, etc.
>
> Also I noticed that, in many figures, test accuracy is plotted along epochs with different hyperparameters. This raises the suspicion that their results were obtained by hyperparameter search against the CIFAR-10 test set. If so, such boost in test accuracy by the proposed method is totally not surprising. Nevertheless, this is just a suspicion, which could be overturned with further evidence.

---

> ### Author Response · Authors · 2023-11-22
>
> Thank you once again,
>
> for clearly communicating your thoughts on the matter. Below we address your concerns.
>
>
>
> **Regarding CMD as a regularizer**
>
> The reference weight in each cluster is chosen as the one which is most correlated to all other members of the cluster (Section 3.2, step 1). This is the cluster’s center in the sense of cosine distance and has similar properties as its L2 fit. Thus we obtain a smooth temporal function, See Fig. 7 (Appendix A.1). Since all members of the clusters are modeled by the reference (up to a multiplicative and additive constant), they all exhibit smooth temporal behavior. Thus, noise caused by the SGD process is strongly removed. This temporal smoothness can alternatively be obtained directly through (weighted) averaging, as done in EMA, SWA. In those works they have already shown that temporal smoothness of the parameters improves performance - on train and test sets alike. Our CMD algorithm achieves similar regularization effects by construction. Thus, while our model only approximates the SGD process, it maintains the essential dynamics in a smoother form, which leads to overall improvement in accuracy.
>
> To support the claim that the regularization effect of CMD is similar by nature to EMA and SWA, we experimented with performing one on top of the other. For instance, EMA was used to regularize CMD modeled trajectories. Indeed, EMA-regularized CMD behaves the same as regular CMD in terms of performance, see Table 8, Appendix D.
>
> **Regarding other Classification datasets**
>
> In response to the reviewer's suggestion, and given the time constraints, we conducted an additional experiment on CIFAR100 using ResNet18, as well as added StarGAN-v2 results. Both exhibit a boost in performance compared to baseline regular training, in terms of accuracy (CIFAR100) and FID and LPIPS (CelebA-HQ). The StarGAN-v2 results, Fig. 15, now offer 4 quantitative graphs and 3 qualitative tables of generated images for both reference image and latent noise guided generation. Notably, this complex setting on high-quality image data demonstrates good generalizability to applicable settings. The CIFAR100 classification results are detailed in Table 7, Appendix D, and demonstrate the effectiveness of both Online CMD and Embedded CMD on this dataset, further validating our method (as well as the inherent boost in performance) beyond CIFAR-10. We recognize the importance of exploring a diverse range of cases. Our manuscript primarily encompasses a wide array of architectures and exhibits CMD abilities on different tasks.
>
>  **Regarding the hyperparameters**
>
> We have analyses addressing such concerns in our manuscript. The primary hyperparameters of CMD are M (number of modes), K (number of sampled weights), and the length of the warm-up phase. These aspects are explored as follows:
>
> Number of Modes (M): An experiment on the impact of varying M is presented in Appendix B, with results shown in Fig. 12. Number of Sampled Weights (K): The effects of different values of K are detailed in Fig. 13, also in Appendix B. Length of Warm-Up Phase: A new experiment, which was conducted following a discussion with the reviewer. We examined the impact of varying warm-up phase lengths, with findings presented in Fig. 16, Appendix C.5.
>
> Regarding the training process hyperparameters (like learning rate, momentum), for each model in Table 1, we used models from various online repositories. The hyperparameters follow their respective repository. Links to these original repositories are provided in Appendix D. In some experiments of the ablation study with ResNet18 (Appendix B), we limited training to 100 epochs, instead of 150 as in Table 1, for time efficiency.

---

> > ### Comment · Reviewer_yr3M · 2023-11-23
> >
> > Thank you for additional clarification.
> >
> > > Regarding CMD as a regularizer
> >
> > I wonder why the authors do not provide any direct evidence for the regularization effect, but rather stick to heuristic explanations or discuss the relationship between SWE and EMA.
> >
> > > Regarding other Classification datasets
> >
> > CIFAR-100 is also a toy dataset that is essentially same as CIFAR-10 in their origin, and thus it hardly strengthens their evidence. Caltech256 and Oxford Flower, which I suggested as examples, are also small-scale but more realistic datasets where the authors could perform the experiments as cheaply as CIFAR-100.
> >
> > > Regarding the hyperparameters
> >
> > The authors' explanation does not resolve my concern about hyperparameter search on the test set. Moreover, I noticed that their supplementary material is now removed from their submission (which was previously available in the initial submission if I remember correctly), so I cannot check whether they did that or not.
> >
> > Since my main concerns remain unresolved, I would like to keep my score.

---

> > > ### Author Response · Authors · 2023-11-23
> > >
> > > **Update: Tiny Imagenet**
> > > We have successfully performed the training of ResNet18 over 75 epochs using Tiny Imagenet, a subset of ImageNet consisting of 200 classes. While our original intent was to train on the entire ImageNet dataset, practical time constraints led us to concentrate our efforts on Tiny Imagenet during this phase of our research. In Fig. 24, we present the test accuracy over epochs, illustrating a carry over of our findings to real-world data. Nevertheless, we recognize that further experimentation, extended training durations, and stabilization of the training process hold the potential to yield more definitive results. In the forthcoming camera-ready version of our paper, we are committed to conducting extensive experiments on this dataset. These experiments will include the presentation of results in a comprehensive multi-training table format, which will incorporate mean and standard deviation statistics for thorough analysis.
> > >
> > > **CMD Regularization**
> > > Figure 7 provides a visual representation of how CMD effectively smoothes temporal dynamics.
> > >
> > > **Other Classification Datasets**
> > > As Beyond CIFAR-10 and CIFAR-100, we have conducted a fresh experiment on Tiny Imagenet, which presents a more realistic and demanding dataset environment.
> > >
> > > **Hyperparameters**
> > > We wish to clarify that the supplementary material, containing a link to our official code implementation, has been consistently accessible in the appendix since our initial submission and remains available for reference. We kindly request that you review the three hyperparameter experiments featured therein (reminder: Figs. 12, 13, 16). Additionally, we have included direct links to the sources of the SGD parameters employed in our experiments. This transparent approach ensures the replicability of our research findings and contradicts any claims of hyperparameter search on the test set.
> > >
> > > We genuinely appreciate your valuable feedback, and we firmly believe that these updates and clarifications enhance the accessibility and credibility of our research. If you have any further inquiries or require additional information, please do not hesitate to contact us.

---

### Official Review · Reviewer_8rcX · 2023-10-31

**Soundness:** 3 good
**Presentation:** 2 fair
**Contribution:** 2 fair
**Rating:** 3
**Confidence:** 4

**Summary:**

The paper presents a novel observation and methodology related to the training dynamics of large-scale neural networks. The authors observe that the parameters during the training of neural networks exhibit intrinsic correlations over time. Capitalizing on this observation, they introduce an algorithm called Correlation Mode Decomposition (CMD). CMD clusters the parameter space into groups, termed modes, that display synchronized behavior across epochs. This representation allows CMD to efficiently encapsulate the training dynamics of complex networks like ResNets and Transformers using only a few modes, enhancing test set generalization as a result. An efficient CMD variant is also introduced in the paper, designed to run concurrently with training, and the experiments indicate that CMD surpasses the performance of existing methods in capturing the neural training dynamics.

**Strengths:**

* The paper introduces a novel observation regarding the intrinsic correlations over time of parameters during the training of neural networks. This insight is leveraged to develop a new algorithm, Correlation Mode Decomposition (CMD), which is a creative contribution to the field.

* Despite the complexity of the topic, the paper seems to be structured and articulated in a manner that allows the reader to follow the authors' logic and methodologies.

**Weaknesses:**

* The citation format within the text could be improved for consistency and adherence to academic conventions. Utilizing citation commands like \citet or \citep would enhance the readability and professionalism of the references within the text.

* Figure 2 Analysis: The benefits of the CMD method as depicted in Figure 2 are not evidently clear. In the left plot, it would be helpful to see the results over a more extended range of epochs to ascertain the method's effectiveness over a longer training period.
In the middle plot, there appears to be a visual discrepancy or blur on the right side that might need clarification or correction.

* Algorithm Explanation and Comparison: A more detailed explanation and justification of the CMD algorithm's design and implementation are necessary for a thorough understanding. The comparison seems limited, primarily focusing on P-BFGS. It would be enriching to see comparisons with other relevant methods, such as pruning methods, to gauge the CMD algorithm's relative performance and novelty.

* The presentation of some concepts and terminologies, such as the "modes" in CMD, might be unclear or lack sufficient explanation, making it challenging for readers unfamiliar with the topic.

*  Discussions or demonstrations regarding the scalability of the CMD algorithm when applied to larger or more complex neural network models would be a valuable addition to assess the method's practical applicability.

* The motivation behind applying CMD in federated learning seems a bit unclear and could benefit from a more explicit demonstration or explanation.

**Questions:**

* Could you clarify the visual discrepancy observed in the middle plot of Figure 2? What does the blur on the right side represent?

* Would it be possible to extend the range of epochs shown in the left plot to provide a more comprehensive view of the CMD method's performance over time?

* Could you elaborate on the motivation and rationale behind applying the CMD method in federated learning?

* What considerations were made in choosing the comparative methods and evaluation criteria in this work?

---

> ### Author Response · Authors · 2023-11-15
>
> Hello,
>
> Thank you for your time and valuable comments. Since the questions you've raised align with the weaknesses, we will directly respond to the weaknesses.
>
> **Weakness 1** – We replaced citations with the preferable \citep, when appropriate.
>
> **Weakness 2** – To demonstrate results across an extended range of epochs, we added Fig. 18 in Appendix D. This figure illustrates the test accuracy of a Vit-b-16 model over 90 epochs of CIFAR10 training using regular SGD training and Online CMD, and includes examples with other models (GoogleNet and WideRes28-10). These experiments align with the experimental setting of Table 1, namely Vit-b-16 is initially pre-trained on ImageNet1K [1], which contains a little more than 1.2 million images. Figure 2's left plot showcases Vit-b-16 pre-trained on the JFT-300M dataset [2], which encompasses over 300 million images. Given the advanced starting point of this pre-trained model, the training duration is significantly reduced (compared to Fig. 18 and Table 1), rendering it less noteworthy. Therefore, we only applied post-hoc CMD in this scenario. Details regarding the pre-trained models have been added to the captions of Fig.2 and Table 1, as well as Appendices B,D, where the experimental details are described.
>
> Regarding the right part of the middle plot in Figure 2, please refer to Fig. 15 in Appendix B, which features more examples of generated images and quantitative comparisons as well. This will let you get a better grasp of the difference between the methods, as there are qualitative differences between the generated images. StarGAN-v2 in its original setting requires EMA (a training regularization technique) to obtain its good results. In Figs. 2, 15 we demonstrate that CMD produces regularization that can be used in this context as well. A more thorough analysis of CMD regularization is provided in the context of image classification, in the Results Sec. 5, under “CMD as a regularizer”.
>
> **Weakness 3** –
>
> **CMD Algorithm's Design and Implementation:**
> Additional details and information regarding the CMD design is provided in Appendix C due to space limitations. We present full psuedo-code algorithms (Appendix C.1), explain how the sampled weights are chosen (Appendix C.2), discuss different methods to divide the model weights to modes (Appendix C.3), etc. Additional references to Appendix C have been added to the methods section to ensure that this information is not overlooked.
>
> **Comparison with Other Methods:**
>    Our comparison primarily focuses on P-BFGS, which is the SOTA in low-dimensional training dynamics. This focus is due to the similarities between CMD and P-BFGS in managing training dynamics using a limited number of modes, a departure from earlier methods that utilized thousands. CMD, like P-BFGS, demonstrates exceptional performance with only a few to several dozen modes.
>
> To the best of our knowledge, P-BFGS, published in IEEE Transactions on Pattern Analysis and Machine Intelligence (March 2023), was the first to showcase such performance. Prior to P-BFGS, there were no direct competitors in such extremely  low-dimensional framework. Additionally, to the best of our knowledge, we are the first to propose such modeling for applications in segmentation and generative models, which our work has successfully demonstrated. Other comparisons: We also compare CMD with known regularization techniques like Exponential Moving Average (EMA) and Stochastic Weight Averaging (SWA). We see limited connection with pruning methods and would appreciate further insight into their suggested relevance.
>
>
> **Weakness 4** – Our work is motivated by previous works related to dynamics analysis (specifically DMD, as mentioned in the Introduction section). Therefore, the nomenclature used in our work is taken from the dynamics analysis area. Particularly, the term 'modes' in our work refers to highly correlated groups of model parameters, as described in Hypothesis 1 in Section 3.1.
>
>
> [1] https://arxiv.org/pdf/1707.02968v2.pdf
>
> [2] https://www.image-net.org/challenges/LSVRC/index.php

---

> ### Author Response · Authors · 2023-11-15
>
> **Weakness 5** – Our work encompasses a range of architectures, demonstrating CMD on larger and complex models as well. This includes smaller models like LeNet-5 (less than 1M parameters), mid-scale ones such as ResNets (up to 36M parameters), and larger frameworks like ViT-b-16 (86M parameters), demonstrating consistency across scales. The CMD parameters used for each experiment (Table 6, Appendix D) show that three model scales (LeNet-5, ResNet18 and ViT-b-16) use the exact same hyper-parameters. We also experiment with the complex StarGAN-v2 architecture (which is a composition of a Generator, a Style Encoder, and a Latent-Noise Mapping network), demonstrating our engagement with large-scale and challenging architectures. These support CMD as a practical method relevant for complex models at scale.
>
> **Weakness 6** – It is true that the motivation should be stated more explicitly, thus we added to Sec. 4’s opening paragraph: “in distributed scenarios, modeling trajectories may reduce the No. of communicated parameters as well, offering further significant benefits. Specifically in Federated Learning (FL), communication efficiency is crucial (Li et al., 2020). For example, Chen et al. (2021) find in their FL experiments that communication accounts for over 50% of the total training time.”

---

> ### Author Response · Authors · 2023-11-22
>
> We would be happy to receive your take on our detailed response. We would like to address any additional issues which are still open before the deadline. In case you feel we have answered most of your concerns we’ll appreciate it if you can re-evaluate positively the overall paper rating. Thanks.

---

### Official Review · Reviewer_t9P8 · 2023-10-31

**Soundness:** 2 fair
**Presentation:** 2 fair
**Contribution:** 3 good
**Rating:** 8
**Confidence:** 3

**Summary:**

This manuscript introduces an interesting idea, namely, *correlation mode decomposition (CMD)*, to cluster the parameter into groups. Instead of considering the top eigenvectors of the Hessian, the idea of CMD efficiently models training dynamics. The insights of CMD can be applied to communication-efficient distributed training.

**Strengths:**

* In general this manuscript is well-structured.
* This manuscript considers an interesting aspect of modeling the training dynamics of complex networks. The idea of using clustered parameters looks novel to the reviewer.
* The manuscript has a good logic flow, from the definition of the post-hoc CMD to online CMD and embedded CMD.
* Sufficient numerical results also justify the effectiveness of the CMD. An extension to FL is also provided in the manuscript.

**Weaknesses:**

* Authors are encouraged to improve the writing quality of the current manuscript.
* Regarding the experiments on FL, it remains unclear to the reviewer why only two communication-efficient FL baselines, namely APF and A-APF, are considered for the evaluation. More recent SOTA methods need to be taken into account.

**Questions:**

NA

---

> ### Author Response · Authors · 2023-11-15
>
> Hello,
>
> Thank you for your time and valuable comments.
>
> **Regarding the writing quality:** We made several adjustments throughout the manuscript following the comments here. We are opened for further suggestions.
>
> For improved readability, an opening paragraph was added to Sec. 3 (Method) so that the reader anticipates the flow mentioned in Strength 3. The opening paragraph of Sec. 4 (CMD for Efficient Communication in Distributed Learning) was added motivation. In Appendix H (which is referenced in the Introduction section 1) we added discussion on: commonalities and differences of DMD and CMD. To better navigate the appendix we added references to Appendix C for details on CMD design, Table 6 for hyper parameters used in the classification models, and Table 4 in the Appendix for performance comparison between online and post-hoc performance.
> Small fixes were introduced: added axis labels to the right plots of Fig. 2, improved Table column titles from “(regular) training” to “baseline SGD”, citations have been improved using \citep, additionally to slight rephrasing and editing.
>
> **We will also note that additional experiments were added:** Fig. 6 in Appendix A.1 visualizes original dynamics alongside their representation, Fig. 16 in Appendix C.5. compares online CMD performance for varying warm-up durations, Fig. 18 in Appendix D provides accuracy graphs for cases from Table 1.
>
> **Regarding Federated Learning (FL):** We acknowledge these are preliminary results, as emphasized in the abstract. Nonetheless, the domain (low-dim. representation of training dynamics) lacks practical examples, thus it was important for us to show this idea, which is also timely. While we plan to further research this direction with CMD, the efficiency of saving communicated parameters via low-dimensional dynamics modeling is not restricted to CMD. We hope that as this domain will progress, other approaches may take inspiration from our proposal, and use their techniques for FL as well.

---

### Meta-Review · Area_Chair_DYAU · 2023-12-07

**Metareview:**

This paper presents an innovative dimension reduction technique, Correlation Mode Decomposition (CMD), to effectively model the learning dynamics of neural networks. Reviewers have positively acknowledged the paper's structure and content, highlighting its methodological clarity and potential impact. Authors addressed concerns raised by reviewers, including detailed explanations and additional experiments, effectively clarifying concerns about experimental design and hyper-parameter selection. Although some skepticism remains among reviewers regarding certain aspects, the authors' comprehensive responses have significantly addressed these issues. Therefore, considering the novel contribution, thorough responses to reviewer feedback, and the potential of CMD in neural network training dynamics, I recommend accepting this paper.

**Justification For Why Not Higher Score:**

The paper, while innovative, maintains the score that reflects its current standing due to limitations in experimental parts. Although the authors have addressed most concerns raised by reviewers, there remain aspects where further validation or comparison with contemporary methodologies could strengthen the paper's claims. This room for enhancement, particularly in experimental design and broader applicability, justifies the decision not to award a higher score.

**Justification For Why Not Lower Score:**

The decision to not assign a lower score is grounded in the paper's clear methodology, innovative approach, and significant potential impact about the problem of neural network training dynamics. The authors have effectively addressed key concerns raised by reviewers, demonstrating a robust understanding of their methodology and its implications. Despite some remaining skepticism, the paper's strengths in innovation and thorough response to feedback make a compelling case for its current score.

---

### Decision · Program_Chairs · 2024-01-16

Accept (poster)